# The influence of mid-latitude cyclones on European background surface ozone

K. Emma Knowland[1,2], Ruth M. Doherty[3], Kevin I. Hodges[4], and Lesley E. Ott[2]

[1]Universities Space Research Association (USRA)/Goddard Earth Science Technology & Research (GESTAR)
[2]Global Modeling and Assimilation Office (GMAO), NASA Goddard Space Flight Center (GSFC), Greenbelt, Maryland, USA
[3]School of Geosciences, University of Edinburgh, Edinburgh, UK
[4]Dept. of Meteorology, University of Reading, Reading, UK

*Correspondence to:* K. Emma Knowland (k.e.knowland@nasa.gov)

**Abstract.** The relationship between springtime mid-latitude cyclones and background ozone ($O_3$) is explored using a combination of observational and reanalysis data sets. First, the relationship between surface $O_3$ observations at two rural monitoring sites on the west coast of Europe – Mace Head, Ireland and Monte Velho, Portugal – and cyclone track frequency in the surrounding regions is examined. Second, detailed case study examination of four individual mid-latitude cyclones and the influence of the associated frontal passage on surface $O_3$ is performed. Cyclone tracks have a greater influence on the $O_3$ measurements at the more northern coastal European station, Mace Head, located within the main North Atlantic (NA) storm track. In particular, when cyclones track north of $53°$ N, there is a significant relationship with high levels of surface $O_3$ ($>$ 75th percentile). The further away a cyclone is from the NA storm track, the more likely it will be associated with both high and low ($<$ 25th percentile) levels of $O_3$ at the observation site during the cyclone's life cycle. The results of the four case studies demonstrate a) the importance of the passage of a cyclone's cold front in relation to surface $O_3$ measurements, b) the ability of mid-latitude cyclones to bring down high levels of $O_3$ from the stratosphere and c) that accompanying surface high pressure systems and their associated transport pathways play an important role in the temporal variability of surface $O_3$. The main source of high $O_3$ to these two sites in springtime is from the stratosphere, either from direct injection into the cyclone or associated with aged airstreams from decaying downstream cyclones that can become entrained and descend toward the surface within new cyclones over the NA region.

## 1 Introduction

Tropospheric ozone ($O_3$), which is harmful to human health and vegetation (Scherrer et al., 2006; Krzyzanowski and Cohen, 2008), has both natural and man-made sources. In the free troposphere, $O_3$ can be transported over large distances as it has a lifetime on the order of a few weeks (Akimoto, 2003; Monks et al., 2009). Rural monitoring ground stations, which are not influenced by local emission sources, are used to monitor the "clean" hemispheric baseline, or background, levels of pollutants such as $O_3$ (e.g., Parrish et al., 2012; Torseth et al., 2012; Wilson et al., 2012). The measurements from these remote stations in Europe previously have been used to study the concentrations of $O_3$ in polluted air arriving from different source regions

separate from the background $O_3$ levels (e.g., Li et al., 2002; Wilson et al., 2012). A large number of observations (about 70 %) taken at the Mace Head station on the west coast of Ireland have been found to be associated with maritime air from over the North Atlantic (NA) and the remainder are associated with continental European air (Jennings et al., 1991; Simmonds et al., 1997).

The meteorological factors – wind speed and direction, humidity and temperature – that impact the variability of pollutant concentrations are determined by the large-scale nearby atmospheric pressure centers. For Western Europe, this is mainly the location of two climatological features: 1) the semi-permanent Icelandic Low between $60°–65°$ N, due in part to the NA storm track (the region known for frequent, strong cyclones) and 2) the semi-permanent Azores High over the Atlantic Ocean near $40°$ N. Studies of airflow relative to cyclones' frontal zones have led to three-dimensional (3D) conceptual models of cyclones

with generally three main airstreams: a warm conveyor belt (WCB; Harrold, 1973), a cold conveyor belt (CCB; Carlson, 1980) and the dry intrusion (DI; Browning, 1997). The horizontal and vertical transport of air within these airstreams can play an important role in the redistribution not only of moisture, heat, and momentum, but also chemical properties of the air from different source regions (Bethan et al., 1998). The frequent cyclones associated with the Icelandic Low can entrain polluted air from different source regions, primarily, Europe and North America (Merrill, 1994; Parrish et al., 2012). During stagnant

conditions, there is pollution build-up which can be dispersed by passing cyclones (Mickley et al., 2004; Hegarty et al., 2007; Leibensperger et al., 2008; Barnes and Fiore, 2013). Within the cyclones, the polluted air at the surface is lofted up into the free troposphere by the WCB (e.g., Bethan et al., 1998; Cooper et al., 2002b; Hannan et al., 2003; Trickl et al., 2003). Once in the free troposphere, pollutants can be transported rapidly in the stronger westerly winds and eventually become mixed with the background composition (e.g., Trickl et al., 2003; Cooper et al., 2004b). The pollutant-enriched air can descend in downwind

systems as part of the DI airstream behind the cold front of a mid-latitude cyclone (hereafter, referred to as cyclone) or as part of a nearby anticyclone (Danielsen, 1980; Bader et al., 1995; Trickl et al., 2003).

In addition to the transport of anthropogenic $O_3$, cyclones enable the vertical transport of $O_3$ from the lower stratosphere into the troposphere (stratosphere-to-troposphere transport: STT) (e.g., Danielsen, 1968, 1980; Sprenger and Wernli, 2003; Škerlak et al., 2014). Several studies have shown that STT events which reach the surface can lead to higher surface $O_3$, of the order of

20-50 ppbv higher than the background levels for the hourly mean, leading to peak surface $O_3$ concentrations of up to 100 ppbv (e.g., Derwent et al., 1978; Davies and Schuepbach, 1994; Lin et al., 2012a). This can result in surface $O_3$ exceeding air quality standards, especially at high elevations such as in the western USA (e.g., Langford et al., 2009, 2015; Lefohn et al., 2012; Lin et al., 2012a, 2014). In particular, Lin et al. (2012a) found that in spring 2010 up to 60 % of total modelled surface $O_3$ in the western USA during air quality exceedances could be attributed to stratospheric intrusions of $O_3$, whilst at Mace Head, using a

tagged tracer approach, Derwent et al. (2004) diagnosed stratospheric $O_3$ to contribute to at least 25 % of the observed annual mean $O_3$. However, the direct connection between STT and peaks in ground-level $O_3$ observations are generally infrequent, with only a small fraction of STT trajectories descending below the mid-troposphere (Viezee et al., 1983; Derwent et al., 1998; Eisele et al., 1999; Stohl, 2001; Škerlak et al., 2014). Air from the free troposphere is largely limited to daytime entrainment into the lowest layer of the atmosphere as the planetary boundary layer (PBL) height increases (e.g., Itoh and Narazaki, 2016;

Ott et al., 2016); however, the ability for $O_3$-rich air to reach the surface depends on a complex array of factors including the

diurnal cycle (Itoh and Narazaki, 2016; Langford et al., 2009, 2012; Ott et al., 2016) and the seasonal cycle of the PBL height (Langford et al., 2015, 2017), the presence of convective mixing (Thompson et al., 1994; Eisele et al., 1999; Langford et al., 2017), and the elevation of the monitoring station, which if located within the free troposphere, especially with the night-time collapse of the PBL, can experience direct STT (Langford et al., 2015, 2017). The influence of the stratospheric air, although weaker by the time the air reaches the surface (Monks, 2000), can last longer in the mid- to lower troposphere if the $O_3$-rich air becomes entrained in high pressure systems behind the surface cold front and continues to subside (Danielsen, 1980; Davies and Schuepbach, 1994; Cooper et al., 2004b).

Previous studies of $O_3$ measurements at these stations have not focused on the specific role of the cyclones on the temporal variability of $O_3$ (e.g., Simmonds et al., 1997; Derwent et al., 2004, 2008). The aim of this study is to quantify the influence synoptic-scale extra-tropical cyclones have on the temporal variability of surface $O_3$ on the west coast of Europe when cyclones are nearby. In particular, we address the following questions:

– Do passing cyclones have a detectable impact on surface $O_3$ observations?

– What are the key synoptic features associated with cyclones that impact surface $O_3$ concentrations?

To address these questions, the sensitivity of $O_3$ measurements at two coastal observation sites – one within the main NA storm track region (Mace Head) and the other south of the main NA storm track region (Monte Velho, Portugal) – will be tested in relation to the frequency and passage of cyclones. This will be followed by detailed analysis of the dynamical processes within four individual cyclones and their possible impact on tropospheric $O_3$ distributions. The focus of this study will be on surface $O_3$ in the spring (MAM) since this is the principal season for intercontinental transport of $O_3$ across the NA (Monks, 2000; Stohl et al., 2002; Dentener et al., 2010). Furthermore, STT events exhibit a springtime maximum due to the build-up of $O_3$ in the lower stratosphere over the winter and spring (Danielsen and Mohnen, 1977; Holton et al., 1995; Monks, 2000; Pausata et al., 2012). Also in springtime, there is a peak in the seasonal $O_3$ cycle at marine, background stations including Mace Head (Simmonds et al., 1997; Wilson et al., 2012; Derwent et al., 2016); at such stations there is often a minimum in $O_3$ in summer as a result of enhanced $O_3$ loss by increased water vapor within the stable marine boundary layer (MBL) (Ayers et al., 1992; Oltmans and Levy, 1994), in contrast to urban environments with the peak in $O_3$ in summer associated with photochemical production. Parrish et al. (2016) and Derwent et al. (2016) demonstrate that the seasonal cycle of $O_3$ at MBL stations, including Mace Head, can be reasonably represented by two harmonics of the seasonal cycle with the first explaining most of the seasonal variation, capturing the summertime $O_3$ loss (and therefore a late winter/early spring maxima) and the spring maxima in $O_3$ within the free troposphere which is entrained into the MBL.

The article continues with a description of the data used, including the $O_3$ observations and the reanalysis variables (Sect. 2). The methodology is presented in Sect. 3. The relationship between cyclones and $O_3$ observations is quantified in Sect. 4. Section 4 also provides a detailed analysis of the airstreams and $O_3$ distributions within the cyclone case studies. Discussion of how cyclones may influence the temporal variability of the surface $O_3$ observations is given in Sect. 5, with final conclusions in Sect. 6.

## 2  Data

Three different types of data are used in this study. The ground-based observation data is first presented followed by descriptions of the reanalyses and a modelled stratospheric tracer.

### 2.1  Surface O$_3$ data

5    Mace Head (MH: 53.3$^\circ$ N, 9.9$^\circ$ W; location indicated by the blue dot on Fig. 1) is located on the west coast of Ireland. Surface O$_3$ measurements, supported by the European Monitoring and Evaluation Programme (EMEP; www.emep.int), are hourly averages of O$_3$ taken every 10 seconds using a commercial UV spectrometer (Model 8810, Monitor Labs, USA) (Derwent et al., 1998; Simmonds et al., 2004) for the period 1 April 1988 – 31 December 2012 (http://actris.nilu.no, last retrieved 6 February 2015).

10    Monte Velho (MV: 38.08$^\circ$ N, 8.8$^\circ$ W; location indicated by the orange dot on Fig. 1) is a rural observation site on the west coast of Portugal, also supported by EMEP, which began hourly surface O$_3$ observations on 1 January 1988 and continued for 21 years, ending on 31 December 2009 (http://actris.nilu.no, last retrieved 6 February 2015). Observations were made using Dasibi analyzer instruments: the original instrument was replaced in 1999 which again failed in 2010 and was not replaced until 2014 (Lourdes Bugalho, pers. comm.). The total number of years in this study for Monte Velho is 19 years, from 1989–2009, 15    exclusive of 1999 and 2005 when data are missing.

### 2.2  Modelled data

Numerous previous studies of long-range transport of trace gases have used chemistry-transport models (CTMs) nudged by reanalysis meteorological fields and are validated against observations. Knowland et al. (2015) was the first study to use the Monitoring Atmospheric Composition and Climate (MACC) project's reanalysis data set (Inness et al., 2013), a combined 20    3D meteorological and composition data set, to explore the relationship between cyclones and air pollution transport of O$_3$ and carbon monoxide (CO). Through a composite cyclone analysis, it was clearly shown that intense cyclones over the North Pacific (NP) and NA regions redistribute O$_3$ and CO horizontally and vertically within the WCB and DI airstreams (Knowland et al., 2015). Therefore, this reanalysis data set can be used to both establish the location and intensity of the cyclones and link the detailed meteorological fields with tropospheric and lower stratospheric composition.

25    In this study, four different model data sets are used which includes three reanalyses – the European Centre for Medium-Range Weather Forecasts' (ECMWF) ERA-Interim reanalysis (Dee et al., 2011), the MACC reanalysis, and NASA's Modern-Era Retrospective Analysis for Research and Applications Version-2 (MERRA-2) reanalysis (Bosilovich et al., 2015) – and a stratospheric tracer data set (Ott et al., 2016). The ERA-Interim cyclone tracks are used here to establish the long-term relationship between cyclones and O$_3$ observations. Since the MACC reanalysis uses a similar atmospheric model to ERA-30    Interim, MACC is used to explore the mechanisms within the case study cyclones that can influence surface O$_3$ concentrations at Mace Head and Monte Velho. The MERRA-2 reanalysis also provides 3D distributions of O$_3$, although less ideal for analysis of surface O$_3$ concentrations since MERRA-2 O$_3$ does not have a detailed chemistry scheme or emission sources for

the troposphere (Wargan et al., 2015, 2017). The MERRA-2 reanalysis, which has the potential to identify more features within the cyclones as the resolution is higher than the MACC reanalysis, is used in conjunction with the MACC reanalysis to provide a measure of uncertainty to the case study analysis. The stratospheric tracer, which uses a similar atmospheric model to the MERRA-2 reanalysis, provides additional information on air mass origin. While models and reanalyses with coarse horizontal resolution ($> 100$ km) are able to identify stratospheric intrusions (Roelofs et al., 2003; Lin et al., 2015; Ott et al., 2016), the fine-scale nature of the $O_3$ filaments is best identified at higher horizontal resolutions (Büker et al., 2005; Lin et al., 2012a; Ott et al., 2016), such as the MACC and MERRA-2 reanalyses and by the stratospheric tracer. However, even high-resolution reanalyses may struggle to represent the complex structure of stratospheric intrusions (e.g. Trickl et al., 2016).

### 2.2.1 The ERA-Interim reanalysis data set

The ERA-Interim reanalysis data set is available from 1 January 1979 to three months prior to present-day at 6-hourly resolution (Dee et al., 2011). It is produced with the ECMWF's Integrated Forecast System (IFS) cycle 31r2 (a full history of changes made to the ECMWF IFS, as indicated by the cycle number, is available at https://www.ecmwf.int/en/forecasts/documentation-and-support/changes-ecmwf-model), with a sequential four-dimensional variational data assimilation (4D-Var) scheme using 12-hourly analysis cycles (Dee et al., 2011). ERA-Interim has a spectral spatial resolution of TL255 (nominally $0.7°$ longitude and latitude) with 60 vertical levels from the surface to 0.1 hPa.

This study uses the ERA-Interim 6-hourly 850 hPa relative vorticity ($\zeta_{850}$) field for cyclone tracking. It also uses the potential vorticity (PV) field for diagnosing the dynamical tropopause whereby the dynamical tropopause is determined as the 2 PVU (1 PV unit (PVU) = $10^{-6}$ K m$^2$ kg$^{-1}$ s$^{-1}$ (Holton et al., 1995)) isosurface.

### 2.2.2 The MACC reanalysis data set

The MACC reanalysis data set, which covers the period 2003–2012, is produced with the ECMWF IFS model two-way coupled to a CTM. The IFS assimilates the same conventional meteorological observations as ERA-Interim as well as satellite observations of three reactive gas species: CO, $O_3$, and nitrogen oxides ($NO_x$) (Flemming et al., 2009; Inness et al., 2013). Although the IFS cycle (cycle 36r1) is a more recent cycle than used for ERA-Interim, the meteorology in the two reanalyses is similar. The CTM is a modified version (Stein, 2009; Stein et al., 2012) of the Model for OZone And Related chemical Tracers (MOZART-3; Kinnison et al., 2007) with the chemical reactions of 115 species in the troposphere and stratosphere (Stein, 2009; Stein et al., 2012; Inness et al., 2013). The two models are run in parallel and, every hour, the IFS provides the CTM with the updated mixing ratios after the horizontal and vertical advection of the assimilated chemical species (Flemming et al., 2009; Stein, 2009; Stein et al., 2012). In turn the CTM provides the IFS system with updated chemical tendencies which are used to constrain the assimilated species. The MACC reanalysis has the same spatial (TL255) and temporal (6-hourly) resolution as the ERA-Interim reanalysis.

In the model evaluation, Inness et al. (2013) compared EMEP $O_3$ observations from all available EMEP monitoring stations at low altitudes ($< 600$ m) to the MACC reanalysis $O_3$ and found that the $O_3$ was underestimated in the reanalysis during spring over central and northern Europe (based on 27 stations within $40°$–$50°$ N and 72 stations within $50°$–$70°$ N, respectively,

absolute monthly biases in spring ranged between about -2 to -9 ppbv for both regions) and agreed well (small overestimation) for southern Europe (based on 5 stations within 30°–40° N, absolute monthly biases in spring ranged between about 1 to 2 ppbv). The reason(s) for the biases in the seasonal $O_3$ is not known although Inness et al. (2013) hypothesize that the biases in the MACC surface $O_3$ are related to the diurnal cycle possibly due to 1) there is no diurnal cycle in the $NO_x$ emissions used in

the CTM resulting in negative $O_3$ biases during the day-time and positive $O_3$ biases during the night-time, 2) misrepresentation of vertical mixing between the boundary layer and free troposphere, and/or 3) less observations are available (and therefore not assimilated) at night. For full details on the input emission data sets and model validation see Inness et al. (2013) and references therein.

In order to evaluate the horizontal and vertical distributions of $O_3$ within cyclones, several parameters from the MACC

reanalysis are used following Knowland et al. (2015). These variables include $\zeta_{850}$, mean sea level pressure (MSLP), zonal and meridional wind components ($u$ and $v$, respectively), vertical velocities ($\omega$), temperature ($T$), specific humidity ($q$) and $O_3$ mixing ratios. Variables required on pressure levels were extracted on twelve pressure levels from 1000 to 200 hPa. The equivalent potential temperature, $\theta_e$, is obtained using T and q. In this study, $\omega$ in pressure coordinates (hPa h$^{-1}$) is multiplied by -1 in order to have the sign be positive to indicate ascent and negative to indicate descent. Since the MACC reanalysis

did not archive PV data, instead the ERA-Interim PV data is used to determine the dynamical tropopause as the MACC and ERA-Interim reanalyses are similar in terms of the IFS and data assimilation.

### 2.2.3 The MERRA-2 reanalysis data set

NASA's MERRA-2 reanalysis is the most recent reanalysis product from NASA's Global Modeling and Assimilation Office (Bosilovich et al., 2015). It is a high-resolution data set (0.5° latitude x 0.625° longitude, 72 vertical levels from surface to

0.01 hPa; Bosilovich et al. (2015, 2016); Gelaro et al. (2017)) for the period from 1 January 1980 to within a couple weeks of real time. The MERRA-2 reanalysis is a product of the Goddard Earth Observing Systems Model, Version 5 (GEOS-5) data assimilation system (DAS) version 5.12.4 atmospheric general circulation model (AGCM), with 3D-Var assimilation with the Gridpoint Statistical Interpolation scheme (GSI; Wu et al. (2002); Derber et al. (2003); Purser et al. (2003a, b)) using 6-hourly analysis cycles. The GEOS-5 model includes a simplified ozone chemistry scheme for both the stratosphere and the

troposphere based on monthly-dependent ozone production and loss rates derived from a two-dimensional chemistry model (Stajner et al., 2008; Bosilovich et al., 2015; Wargan et al., 2015). For the period since October 2004, the high-resolution stratospheric $O_3$ profiles from the Microwave Limb Sounder (MLS; Waters et al. (2006)) and total column ozone from the Ozone Mapping Instrument (OMI, Levelt et al. (2006)), both aboard NASA's Aura satellite, are assimilated (Bosilovich et al., 2015; Wargan et al., 2015; McCarty et al., 2016). Since the assimilated $O_3$ observations are mainly stratospheric, it is expected

that the MERRA-2 analysis $O_3$ will realistically represent the direct influence of stratospheric $O_3$ into the troposphere, such as STT events, however $O_3$ concentrations will be biased low elsewhere in the troposphere (Ott et al., 2016).

The MERRA-2 variables used in this study include 6-hourly SLP, $u$, $v$, $\omega$, Ertel's PV (EPV), T, q, and $O_3$ mixing ratios (GMAO, 2015a, b).

#### 2.2.4 GEOS-5 stratospheric tracer

To provide more information on air mass origins, a simple stratospheric tracer has also been implemented in GEOS-5 (stratospheric fraction (STFR); Ott et al. (2016)). At every model time step (7.5 minutes), the tracer is set to 1 above the tropopause (defined as the higher height of the thermal or dynamical tropopause) and reset to 0 at the surface. Values of this tracer quantify
the fraction of air that has recently been in the stratosphere. For this study, STFR was computed by running the GEOS-5 AGCM constrained by the MERRA-2 meteorological reanalysis to ensure physical consistency with the ozone and meteorological reanalysis fields.

## 3   Methods of analysis of high and low $O_3$ observations and their relationship to cyclone tracks

### 3.1   Cyclone identification

The objective feature tracking algorithm, TRACK (Hodges, 1994, 1995, 1999), can be used to track synoptic-scale features such as minima in MSLP or maxima in relative vorticity (e.g., Hoskins and Hodges, 2002; Bengtsson et al., 2006, 2009; Catto et al., 2010; Knowland et al., 2015). Here, TRACK is used to identify cyclones as maxima in $\zeta_{850}$ in the ERA-Interim and MACC reanalyses. To be consistent with previous studies using TRACK, a spectral analysis, based on a spherical harmonic expansion, is performed which allows the large-scale background field to be removed (planetary-scale waves with total wave number, n,
$\leq 5$). At the TL255 resolution of the ERA-Interim and MACC reanalyses, vorticity contains many small-scale features such as frontal regions, shear zones, and mesoscale features; therefore, to focus on the synoptic scales, the resolution is reduced to T42 using triangular truncation and spectral tapering of the coefficients (Hoskins and Hodges, 2002). The tracking is performed on the unit sphere and the tracks are constructed by minimizing a track smoothness cost function subject to adaptive constraints (Hodges, 1995, 1999). On completion of the tracking process, the stationary cyclones (travel < 1000 km during the cyclone's
life cycle) and short-lived cyclones (cyclone's life cycle lasts < two days) are removed to focus on those cyclones which are likely to be mobile (Hodges, 1999) and hence contribute to $O_3$ transport.

#### 3.1.1   $O_3$ observations associated with cyclone tracks

In order to relate the EMEP hourly $O_3$ observations to the cyclone frequencies, the hourly $O_3$ observations were averaged together every six hours, with the first of the six hours corresponding to the reanalysis time step. The 6-hourly averaged $O_3$
data are then matched to the cyclone tracks based on the reanalysis time step. The cyclone tracks, with the accompanying $O_3$ data, are then sorted by different regions relative to the observation site. A cyclone is considered as passing near Mace Head if one 6-hourly cyclone track point occurs within a 20° latitude by 20° longitude region, approximately $3\text{x}10^6 \text{ km}^2$, centered on Mace Head (hereafter referred to as the Center region, 43°–63° N, 20° W–0°, yellow box, Fig. 1a). The size of the region was defined in order to capture the large spatial coverage of each cyclone, and in particular to capture the trailing cold fronts which
can be on the order of 100 to 1000 km (Shapiro and Keyser, 1990). Four additional regions (North, South, East, and West) in the vicinity of Mace Head were defined to test the influence of the location of the cyclones and their associated fronts in

relation to surface $O_3$ at Mace Head (see Fig. 1 caption). The equivalent five regions around Monte Velho (orange dot, Fig. 1) are defined such that the Center region is the 20° by 20° region centered around Monte Velho (28°–48° N, 19° W–1° E), with the other regions defined in the same fashion as around Mace Head (not shown). The larger East and West domains in Fig. 1b are defined to examine the influence of the cyclones potentially interacting with relatively clean maritime air to the west of
5   sites and relatively polluted continental air to the east of sites.

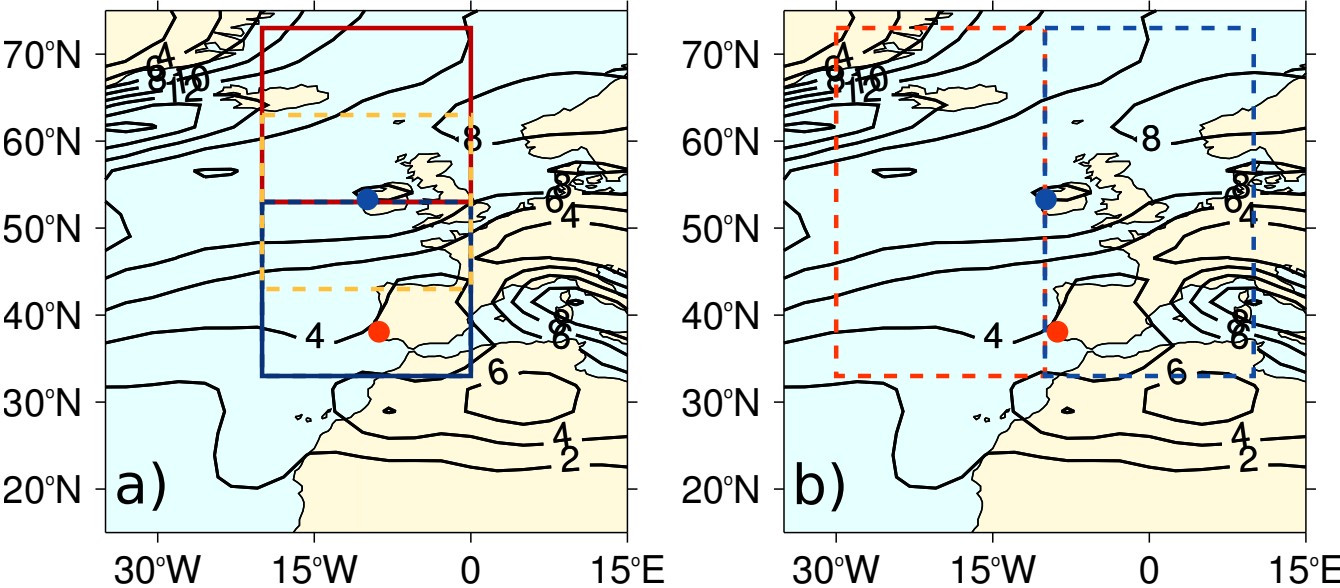

**Figure 1.** Maps showing the five regions where cyclones pass near Mace Head (blue dot) which are associated with $O_3$ measurements from Mace Head: **(a)** North region 53°–73° N, 20° W–0° (red box), Center: 43°–63° N, 20° W–0° (yellow box) and South: 33°–53° N, 20° W–0° (blue box) and **(b)** West: 33°–73° N, 30°–10° W (orange box) and East: 33°–73° N, 10° W–10° E (blue box). In addition, track density distribution for cyclones from ERA-Interim for MAM 1979–2011 (black contour lines; interval of 2 cyclones per month per 5° spherical cap) and Monte Velho (orange dot) are indicated.

For each cyclone track point which occurred within the specified region, the accompanying 6-hourly averaged $O_3$ observation was classified as high, average or low. To perform this analysis, the 6-hourly averaged $O_3$ data are first sorted by magnitude, and the percentile (pc) values were calculated in order to identify high $O_3$ ($> 75^{th}$ pc values) and low $O_3$ ($< 25^{th}$ pc values) for each spring at the two stations. The extreme high and extreme low $O_3$ values ($95^{th}$ and $5^{th}$ pc, respectively), to represent the
10  peak and baseline concentration levels, respectively, based on Wilson et al. (2012), are included in the Supplemental material. The percentile values were calculated for each spring separately to avoid capturing the increasing trend in background $O_3$ over the observation period. If the accompanying 6-hourly averaged $O_3$ observation was classified as either high or low $O_3$ at Mace Head then the entire cyclone track is classified as associated with high or low $O_3$, respectively. However, with this methodology, it is noted that a single cyclone-track can be associated with both $O_3$ criteria.

To test if there is a significant difference in the levels of $O_3$ observed at Mace Head or Monte Velho associated with the passing of a cyclone, the $\chi^2$ contingency test was used (Press et al., 1992). By calculating the $\chi^2$ value for each season over the period of available observations, the $\chi^2$ contingency test shows the number of years where there is a significant relationship between the high and low $O_3$ observations at the surface and the number of cyclone tracks in the area.

### 3.1.2 Case study cyclone identification

To investigate the temporal variability of $O_3$ at the surface with the passing of the cyclones and their associated fronts, several individual cyclones were identified in the MACC reanalysis with the specific objective of determining how high $O_3$ is associated with cyclone passage and dynamics. Over the NA region for the period 2003–2012, there is an exact match of about 90 % of the cyclone tracks between the MACC reanalysis and the ERA-Interim reanalysis (matching criteria similar to Hodges et al. (2011) of a minimum mean separation distance of $5°$ (geodesic) and at least 10 % of the track points overlap in time), with discrepancies generally occurring for weaker cyclones (Knowland, 2016). This is expected since the MACC reanalysis and the ERA-Interim reanalysis use similar systems in terms of data assimilation and IFS model, and assimilate the same meteorological observations.

The following criteria were used to select cyclones for the case studies:

– Cyclones passing through the North and South regions of the two sites associated with high 6-hourly averaged $O_3$ ($> 75^{th}$ pc) were identified.

– Of all the cyclones identified, the strongest cyclones (where maximum $\zeta_{850} > 80^{th}$ pc of springtime NA $\zeta_{850}$ ($8.0 \times 10^{-5}\ \mathrm{s}^{-1}$) within the region) were selected. This ensures that the cyclones will be within the top 20 % of the cyclones in the NA and that the cyclone is an exact match identified in both the ERA-Interim and the MACC reanalysis data sets (not shown). Moreover, the airstreams within strong cyclones will be more evident than in the weaker cyclones (Catto et al., 2010).

The strongest storms then have to satisfy the following additional criteria:

– The cyclones must have two or more 6-hourly time steps associated with high $O_3$. This selects the cyclones that are associated with elevated levels of $O_3$ at the monitoring station which can be related to averaging periods typically used for policy metrics such as the daily maximum 8-hour average $O_3$.

– The difference between the high 6-hourly averaged $O_3$ observations and the MACC $O_3$ at 1000 hPa within the grid box of the observation site is $\leq 20$ %.

If the additional criteria are not satisfied for the strongest cyclone, the criteria are then examined for the next strongest storm and so on until a cyclone is found which satisfies the criteria.

From the total MACC reanalysis cyclone tracks in the North and South regions, there were 171 and 121 cyclones in each region, respectively, that satisfied the requirement of high $O_3$ at Mace Head (Table 1). Of the 171 cyclones associated with high $O_3$ at Mace Head while in the North region, 20 of the cyclones are classified as strong (Table 1). The strongest of the 20

| | Number of cyclones (by strength) associated with high $O_3$ ($O_3 > 75^{th}$ pc) | | | | |
|---|---|---|---|---|---|
| **Mace Head** | All cyclones | Strong cyclones | Moderate cyclones | Total tracks | Percent of all cyclones associated with high $O_3$ |
| North | 171 | 20 (**N1**) | 78 | 319 | 54 % |
| South | 121 | 6 (**S1**) | 32 | 253 | 48 % |
| **Monte Velho** | | | | | |
| North | 95 | 3 (**N2**) | 28 | 213 | 45 % |
| South | 69 | 0 | 4 (**S2**) | 117 | 59 % |

**Table 1.** The number of cyclones by strength (where strong cyclones and moderate cyclones are a subset of all cyclones) from the MACC reanalysis associated with high 6-hourly averaged $O_3$ observations at Mace Head for the period 2003–2012 and at Monte Velho for the period 2003–2009 (exclusive of 2005) using the 6-hourly averaged $O_3$ pc values. The case study cyclones are included in the respective category.

cyclones in the North region is selected for further study and hereafter referred to as the N1 cyclone (Table 1; see Fig. 2a). The N1 cyclone is one of the 58 intense cyclones used in the NA composite analysis of Knowland et al. (2015). In the South region, there are fewer strong cyclones associated with high $O_3$ at Mace Head (6 of the 121 cyclones, Table 1). The strongest cyclone in the South region will be referred to as the S1 cyclone (Table 1; see Fig. 2a). The N1 and S1 cyclones, with 4 and 5 high $O_3$

time steps whilst in the region, respectively, both satisfied the two additional criteria.

The same procedure was performed to select the case study cyclones for the regions near Monte Velho (95 and 69 cyclones in the North and South regions, respectively, Table 1). In the region to the north of Monte Velho, only 3 of the 95 cyclones which are associated with high $O_3$ observed at Monte Velho are classified as strong cyclones (Table 1). The third strongest cyclone was selected for further study as out of the three cyclones it was the only system with persistent high $O_3$ (2 time steps).

This cyclone will be referred to as the N2 cyclone (Table 1; see Fig. 2b).

In the South region, there are no "strong" cyclones associated with high $O_3$ at Monte Velho (Table 1). Here, the $\zeta_{850}$ threshold value was reduced to select moderate cyclones, where the maximum $\zeta_{850} > 50^{th}$ pc of springtime NA $\zeta_{850}$ ($5.83\times10^{-5}$ s$^{-1}$), which identified 4 moderate cyclones in the South region associated with high $O_3$ (Table 1). The strongest of these four cyclones, the S2 cyclone (Table 1; see Fig. 2b), satisfies the additional criteria, in particular, during the S2 life cycle within the

South region, there were 7 time steps associated with high $O_3$ at Monte Velho.

### 3.1.3   The Thermal Front Parameter function

The distribution of $O_3$ within the airstreams of the case study cyclones is examined using both the geographical coordinate system and the cyclone-centered coordinate system (See Appendix Bengtsson et al., 2007). In the geographical coordinates, archived MSLP analysis charts can be used to identify the fronts. In the cyclone-centered coordinate system, the cyclone fronts

can be identified using an objective frontal location parameter. The thermal front parameter (TFP) function, given by Eq. 1, is the frontal location parameter calculated using $\theta_e$ on a given pressure surface (Clarke and Renard, 1966). With knowledge of

the front locations, the airstreams within the strongest cyclones can then be identified and the vertical transport pathways for $O_3$ can be examined.

$$TFP \equiv -\nabla |\nabla \theta_e| \cdot \frac{\nabla \theta_e}{|\nabla \theta_e|} \tag{1}$$

## 4  Results

### 4.1  Overview of the relationship between cyclone frequency and surface $O_3$

The observed $O_3$ levels at the two sites are first examined in relation to the ERA-Interim reanalysis cyclone tracks due to its longer time period to match the $O_3$ observations. Here, tracks were not filtered for intensity. For springtime during the period of observations from 1988–2010 at Mace Head, the average number of the total cyclone tracks associated with $O_3 > 75^{th}$ pc (hereafter, high $O_3$), $O_3 < 25^{th}$ pc (hereafter, low $O_3$) or both high and low $O_3$ is shown in Table 2 as a percentage of the total tracks in each region. In addition, the number of years during 1988–2010 which have more tracks associated with high $O_3$ than with low $O_3$ and the number of years which have more tracks associated with low $O_3$ than with high $O_3$ are shown in Table 2, including the number of years where this difference is significant ("SGF") based on the $\chi^2$ statistic.

There are higher mean percentages of tracks passing through the North and Center regions associated with high $O_3$ at Mace Head, 52 and 51 %, respectively, than with low $O_3$ at Mace Head, 37 and 41 %, respectively (Table 2). The North region has the greatest number of years (18 out of 23 years, Table 2) where the percentage of tracks associated with high $O_3$ is greater than the percentage of tracks associated with low $O_3$, where 15 of these years have a significant $\chi^2$ difference (North region (*SGF*), Table 2). The Center region also has a large number of years (17 years) where the percentage of tracks associated with high $O_3$ is greater than the percentage of tracks associated with low $O_3$, although only about a third of those years have a significant difference (Table 2). In the South region, in contrast to the North and Center regions, there are more cyclones passing south of Mace Head (Table 2) which are associated with low $O_3$ at Mace Head (53 %) than with high $O_3$ (45 %). In the South region, there are 16 years which have more tracks associated with low $O_3$ than with high $O_3$, of which half of these years have significant differences (Table 2). The transition from significantly more cyclones in the North and Center regions (located within the main storm track region, Fig. 1) associated with high $O_3$ at Mace Head to significantly more cyclones in the South region (generally south of the main storm track region, Fig. 1) associated with low $O_3$ at Mace Head illustrates how $O_3$ levels are sensitive to the passage of cyclones and their frontal zones. The percent of tracks which have both high and low $O_3$ increased from 12 % in the North region to 20 % in the South region (Table 2).

The region east of Mace Head shows a slightly higher mean percentage of tracks associated with high $O_3$ compared to low $O_3$ (48 compared to 45 %, Table 2) with 15 out of 23 years marked by more tracks associated with high $O_3$ than low $O_3$ (6 of which were significant, Table 2) and 15 % of the tracks associated with both high and low $O_3$ observations. The West region has only a slightly higher mean percentage of tracks associated with low $O_3$ compared to high $O_3$ (52 compared to 51 %), with about half of the years (12 out of 23 years) with more tracks associated with low $O_3$ than high $O_3$ (Table 2) and 20 % of

| Region | Mean percent of tracks per season associated with $O_3 > 75^{th}$ pc | Number of years with more tracks associated with $O_3 > 75^{th}$ pc (SGF) | Mean percent of tracks per season associated with $O_3 < 25^{th}$ pc | Number of years with more tracks associated with $O_3 < 25^{th}$ pc (SGF) | Mean number of tracks per season (MAM) | Percent of tracks associated with both high and low $O_3$ |
|---|---|---|---|---|---|---|
| **Mace Head** | | | | | | |
| North | 52 % | 18 (15) | 37 % | 5 (0) | 32 | 12 % |
| Center | 51 % | 17 (6) | 41 % | 6 (1) | 30 | 14 % |
| South | 45 % | 7 (2) | 53 % | 16 (8) | 25 | 20 % |
| West | 51 % | 11 (0) | 52 % | 12 (2) | 50 | 20 % |
| East | 48 % | 15 (6) | 45 % | 8 (2) | 52 | 15 % |
| **Monte Velho** | | | | | | |
| North | 51 % | 8 (1) | 52 % | 11 (3) | 29 | 18 % |
| Center | 55 % | 11 (1) | 50 % | 8 (2) | 24 | 23 % |
| South | 61 % | 10 (2) | 61 % | 9 (3) | 17 | 31 % |
| West | 60 % | 7 (4) | 61 % | 12 (2) | 39 | 29 % |
| East | 58 % | 12 (4) | 56 % | 7 (0) | 45 | 26 % |

**Table 2.** The mean percent of cyclone tracks associated with high $O_3$ ($> 75^{th}$ pc) and low $O_3$ ($< 25^{th}$ pc) measured at the surface at **(top)** Mace Head based on each MAM season in the period 1988–2010 and **(bottom)** Monte Velho based on each MAM season in the period 1989–2009 (excluding 1999 and 2005 which had insufficient data). The total number of years in the **(top)** 23-year period (1988–2010) and **(bottom)** 19-year period (1989–2009, excluding 1999 and 2005) which have a greater number of tracks associated with $O_3 > 75^{th}$ pc and $O_3 < 25^{th}$ pc are also given. The total number of those years which have a significant $\chi^2$ difference at $p < 0.05$ confidence interval are given in the brackets (*SGF*). See Sect. 3.1.1 for definition of regions.

the tracks associated with both high and low $O_3$ observations. Although the differences are small and that this method did not test which direction the cyclone tracks passed through the region, these results suggest that the cyclones in the East region are associated with more polluted air than cyclone tracks in the West region.

The relationship between cyclone location and surface $O_3$ is also examined for the southern site, Monte Velho (Table 2). The Center region has a greater mean percentage of cyclone tracks associated with high $O_3$ (55 %) compared to those associated with low $O_3$ (50 %), whilst in the remaining regions the difference between the mean percent of tracks associated with high $O_3$ and with low $O_3$ is small (Table 2). The number of these years having more tracks in the North, Center, and South regions associated with high $O_3$ is almost equal to the number of years which had more tracks associated with low $O_3$. Very few years having a significant $\chi^2$ difference, possibly due to the percent of tracks associated with both high and low $O_3$ increases (up to 31 % of tracks) the further away from the main NA storm track (Table 2). However, there are a greater number of years (12 out of 19 years) where the cyclone tracks are associated with low $O_3$ to the west of Monte Velho, and also 12 years where the cyclone tracks are associated with high $O_3$ when tracks are to the east of Monte Velho (Table 2). Despite the small differences in the mean percent of tracks (and the small number of significant years) associated with high and low $O_3$ at Monte Velho, this indicates that the cyclone tracks still play a role in affecting surface $O_3$ measurements at Monte Velho associated with the temporal variability of $O_3$.

In summary, the $O_3$ measurements at Mace Head and Monte Velho are influenced by cyclone tracks passing in the vicinity of the observation sites. The relationship between the $O_3$ observed and the cyclone tracks varies at the two observation sites. In the following sections, several individual cyclone tracks are selected using the MACC reanalysis and examined using the MACC and MERRA-2 reanalyses and the STFR tracer to identify the possible sources of high $O_3$ as well as the transport pathways of $O_3$ within the cyclones which can affect the surface $O_3$ observations at Mace Head and Monte Velho.

## 4.2   Cyclone case studies associated with high $O_3$ at Mace Head

The four case study cyclones, selected in Sect. 3.1.2, are shown in Fig. 2 with the tracks colored by the corresponding MACC reanalysis $O_3$ at Mace Head (N1 and S1 tracks, Fig. 2a) and Monte Velho (N2 and S2 tracks, Fig. 2b).

The N1 cyclone occurred during March 2007. The N1 cyclone track across the NA is further south than the S1 cyclone track before it turns northward, remaining to the west of Mace Head. During the time the N1 cyclone is in the North region (4–7 March 2007), high $O_3$ ($> 55$ ppbv) occurs for about a day and a half at Mace Head. The S1 cyclone occurred at the end of April 2012. As the S1 cyclone passes to the south of Mace Head, there is an increase in $O_3$ until it leaves the South region.

In May 2006, the N2 cyclone moves through the North region about $10°$ (approximately 1000 km) to the north of Monte Velho (Fig. 2b), so that Monte Velho was likely impacted by the trailing cold front. The N2 cyclone is a good example of how surface $O_3$ levels can vary during the life cycle of a passing cyclone. At the time of maximum vorticity, the surface $O_3$ at Monte Velho is $< 30$ ppbv, while just before the N2 cyclone leaves the North region the $O_3$ concentration at Monte Velho has nearly doubled (Fig. 2b). The S2 cyclone also occurred in March 2007, shortly after the N1 cyclone. The S2 cyclone originates about $10°$ to the west of Monte Velho and slowly tracks to the southeast toward and across North Africa (Fig. 2b). There is high $O_3$ ($> 45$ ppbv) at Monte Velho during the majority of the S2 cyclone's life cycle in the South region.

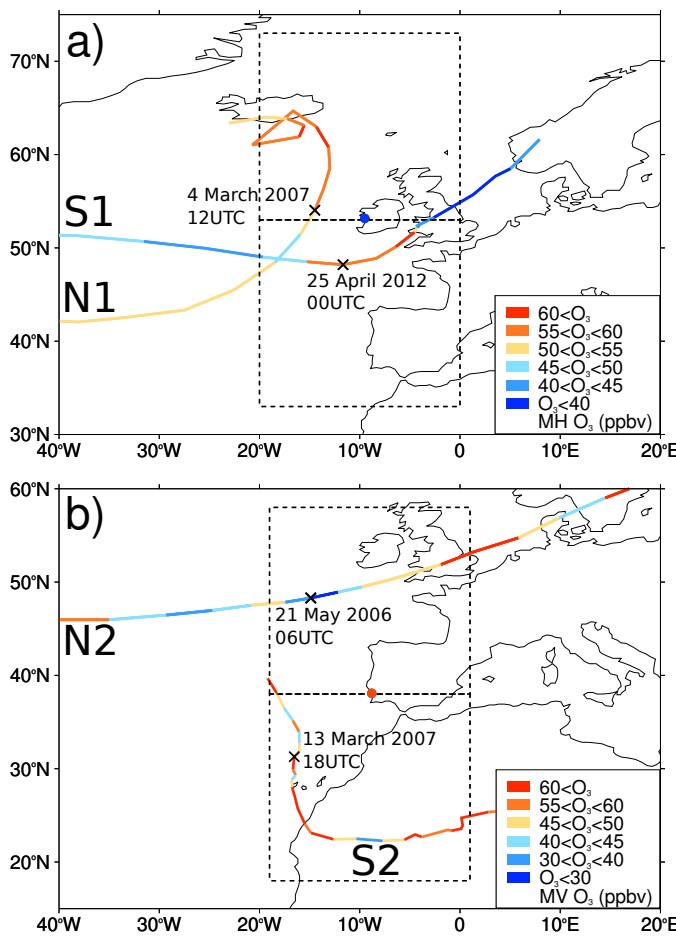

**Figure 2. (a)** Springtime strong cyclone tracks, N1 and S1, associated with high $O_3$ ($O_3 > 75^{th}$ pc) at Mace Head (blue dot) while in the North and South regions, respectively. Tracks colored by MACC $O_3$ interpolated to the Mace Head location. **(b)** Springtime strong cyclone track, N2, in the North region and moderate cyclone track, S2, in the South region which are associated with high $O_3$ at Monte Velho (orange dot). Tracks colored by MACC $O_3$ interpolated to the Monte Velho location. The time of maximum vorticity of each track is given at the location along the track, indicated by the cross.

The following sections will explore in more detail how the dynamics of the case study cyclones influenced the surface $O_3$ at Mace Head and Monte Velho. The synoptic conditions during different stages in the life cycle of the case study cyclones are described using the MACC reanalysis data set, and where stated, also using the MERRA-2 reanalysis data set and the STFR tracer, in order to investigate the importance of time and location of the cyclone relative to the high and low $O_3$ observations at the monitoring stations.

#### 4.2.1 N1 cyclone: synoptic conditions associated with high $O_3$ at Mace Head

The N1 cyclone is a secondary frontal cyclone which formed over the western NA (42.6° N, 50.6° W) on 2 March 2007 06 UTC from the parent low located between Greenland and Iceland. Analysis of the N1's life cycle indicates there is high $O_3$ already in the lower troposphere over the Labrador Sea (up to 70 ppbv at 1000 hPa) which had descended from the stratosphere as part of the DI airstream of the parent low (not shown). This $O_3$-rich air appears to be advected toward Mace Head by the N1 cyclone (Figs. 3a and 4). Nearly two days after the N1 cyclone formed, high levels of $O_3$ (> 55 ppbv, Fig. 3a) at 1000 hPa are then advected southeastward into the NA behind the cold front of the N1 cyclone (see Fig. 3b for location of fronts) and northward around the parent low (Fig. 3a). The high $O_3$ eventually reaches Mace Head on 5 March 2007 at midnight, after the cold front has passed over Mace Head (Fig. 4d), and persists until 6 March 2007 at 12 UTC.

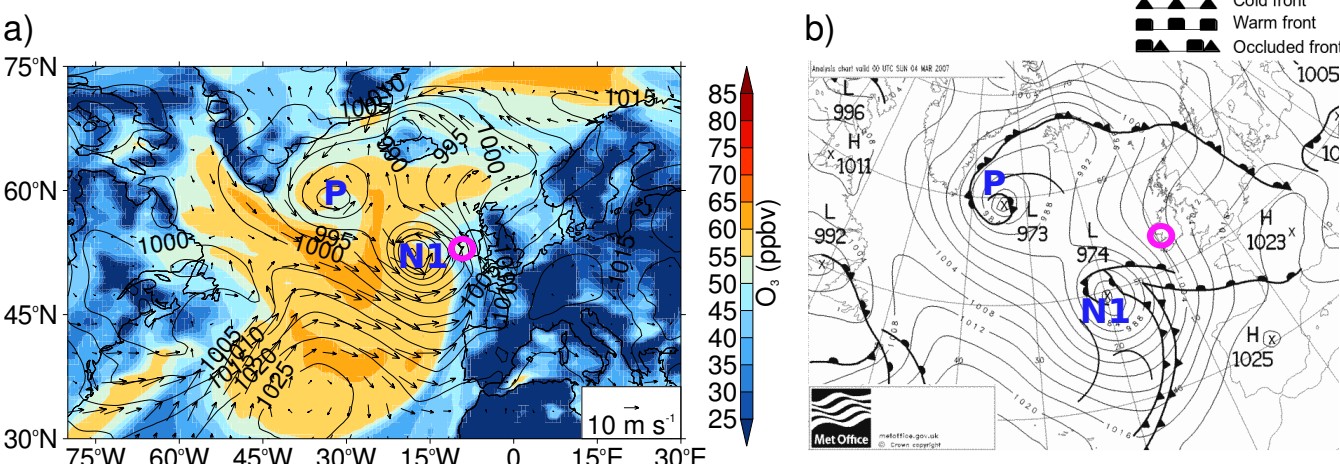

**Figure 3. (a)** 1000 hPa MACC $O_3$ (color) and horizontal wind vectors (10 m s$^{-1}$ reference arrow) and MSLP (solid contours, 5 hPa intervals) on the 4 March 2007 at 00 UTC, 12 hours prior to maximum $\zeta_{850}$ for N1 cyclone (N1: 52° N, 19° W). Parent low (P: 59° N, 34° W) is located between Greenland and Iceland. Mace Head indicated by pink open circle. **(b)** Analysis chart for 4 March 2007 at 00 UTC from UK Met office (available from archive at http://www.wetterzentrale.de/topkarten/tkfaxbraar.htm, retrieved 19 May 2015) shows the complex fronts associated with the N1 cyclone and the parent low. Mace Head (pink circle) is located to the north of the N1 warm front.

At the time of maximum vorticity ($\zeta_{850}$ =11.56x10$^{-5}$ s$^{-1}$, 4 March 2007 12 UTC, not shown), the N1 cyclone and its decaying parent low merged at the surface, intensifying the N1 cyclone. Strong westerly winds developed throughout the troposphere across the NA region as a result of the large pressure difference between the low pressure systems in the NA (the N1 cyclone and an upstream low over Newfoundland) and the Azores High (Fig. 4d). In addition to the descent (white contour lines indicate $\omega$, Fig. 4a-c) of stratospheric $O_3$-rich air within the DI of the N1 cyclone, $O_3$-rich air from aloft (300 and 500 hPa) has descended toward the surface from the stratosphere as part of DI in the upstream low (Fig. 4a-d). The strong westerly winds appear to transport this $O_3$-rich air across the western NA (Fig. 4a-d). This results in persistent high $O_3$ observed at Mace Head

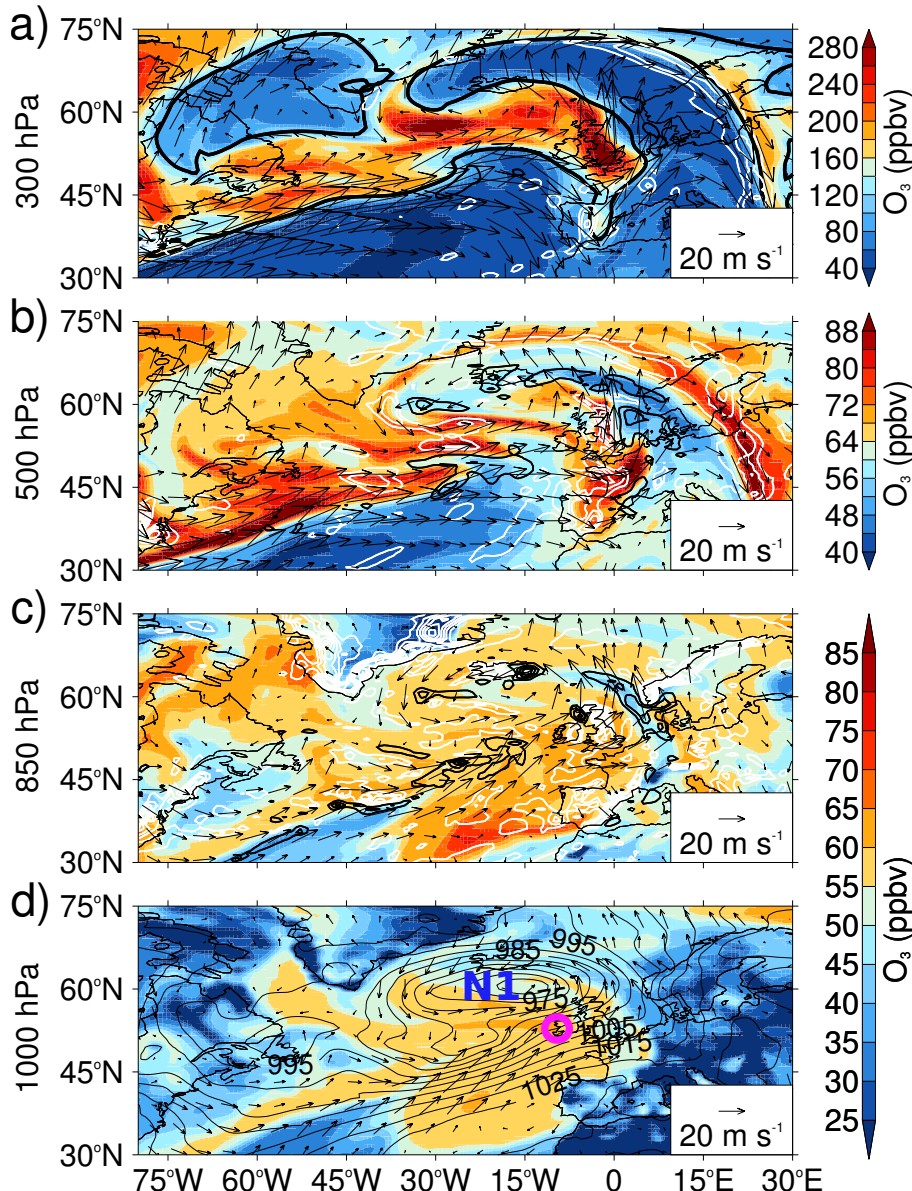

**Figure 4.** Synoptic conditions and O$_3$ distribution for the N1 cyclone (**d**; 59° N, 13° W) on 5 March 2007 at 00 UTC, 24 hours after Fig. 3. High O$_3$ was reported at Mace Head. O$_3$ (color; note different scales used for different levels) and horizontal winds (20 m s$^{-1}$ reference arrow) are shown on four levels: (**a**) 300 hPa, (**b**) 500 hPa, (**c**) 850 hPa, and (**d**) 1000 hPa. In addition, MSLP (**d**; solid contours, 5 hPa intervals), $\omega$ (**a**, **b**, and **c**; black contours for positive values indicating ascent, 15 hPa h$^{-1}$ contour intervals, and white contours for descent, -5 hPa h$^{-1}$ contour intervals), and 2 PVU isosurface (**a**; thick contour) are shown.

in the days following the maximum vorticity of the N1 cyclone (Fig. 4d), and even after the cyclone has left the North region (Fig. 2a).

The N1 cyclone is a good example of how one cyclone can bring both low and high $O_3$ to Mace Head. Prior to maximum vorticity, low $O_3$, defined to be less than the MAM 2007 6-hourly averaged $O_3$ $25^{th}$ pc value of 39.1 ppbv, is reported at Mace Head on 4 March 2007 00 UTC (Fig. 3a). Figure 3a shows there is a sharp contrast in $O_3$ ($> 10$ ppbv) at the cold frontal boundary, with higher $O_3$ levels behind the cold front and lower $O_3$ levels ahead of the cold front in the warm sector between $30°–50°$ N. This low $O_3$ is advected from the subtropics northward towards Mace Head parallel to the cold front within the warm sector (Fig. 3a).

The distribution of $O_3$ within the airstreams of the N1 cyclone can be examined by using a cyclone-centered coordinate system (Figs. 5–7 and S1–S4). Here, the reanalyses are sampled onto a radial grid centered on the cyclone track points within a $20°$ (approximately 2000 km) radius with the cyclones rotated to be moving from left to right in the figures following Bengtsson et al. (2007). For discussion of the cyclone-centered maps, the unit circle with cardinal points are used instead of using geographical coordinates such that the direction of cyclone propagation is from "west" to "east" (Knowland et al., 2015).

In the cyclone-centered coordinate system, the fronts are identified using the TFP function (Fig. 5). In Fig. 5a, the TFP values on the 925 hPa surface on the 4 March 2007 00 UTC show there is a baroclinic zone (strong horizontal temperature gradient on a constant pressure surface) or front from the N1 cyclone center toward the west/southwest (relative to the direction of cyclone propagation) as well as another one to the south/southeast. However, this does not indicate whether the fronts are warm fronts or cold fronts. Using the synoptic chart in Fig. 3b, in combination with the system-relative winds (where the speed of the cyclone propagation has been subtracted from the winds about each cyclone (Catto et al., 2010; Dacre et al., 2012; Knowland et al., 2015)) and the MSLP contours on Fig. 5, the front trailing from the center of the N1 cyclone toward the west is diagnosed as a cold front and the weaker baroclinic zone in the bottom right quadrant is a warm front (Fig. 5a). There is an additional baroclinic zone associated with the parent low in the top right quadrant as well as other baroclinic zones within the warm sector of the cyclone that are not seen in the analysis chart (Fig. 3b). A day later, the N1 cyclone has become occluded and the baroclinic zones are weak around the cyclone center and along the trailing cold front which is now further to the south (Fig. 5b).

The main feature at 4 March 2007 00UTC in both the meteorological fields and $O_3$ distribution is the cold frontal boundary which separates the WCB and DI airstreams (Fig. 6). The strong cold frontal boundary features as the sharp gradient in the 1000 hPa $O_3$ distribution and the curve in the isobars in the southwest quadrant of Fig. 6c (similar to Fig. 3). The front is identified between -5° to -2° throughout the troposphere by a strong gradient in equivalent potential temperature, $\theta_e$, from 1000 to 400 hPa, located between the high values of $\theta_e$ (warm and moist) to the south of the cyclone center (-15° to -5°) compared to the lower values of $\theta_e$ (cool and dry) to the north (-2° to 15°, Fig. 6b). Ahead of the cold front, the WCB airstream transports $O_3$-poor air into the N1 cyclone (Fig. 6). The WCB airstream can be identified by the region of strong ascent (Fig. 6a) throughout the troposphere (-5°, 1000–400 hPa; Fig. 6b), with the maximum ascent to the north of Mace Head (maximum $\omega$ $> 60$ hPa h$^{-1}$ at 700 hPa, Fig. 6a,d). Within the N1 cyclone, $O_3$-rich air descended toward the surface within the DI airstream, however this is far from Mace Head (about 1400 km) at this time (Fig 6a,b). Stratospheric $O_3$-rich air, identified here by high

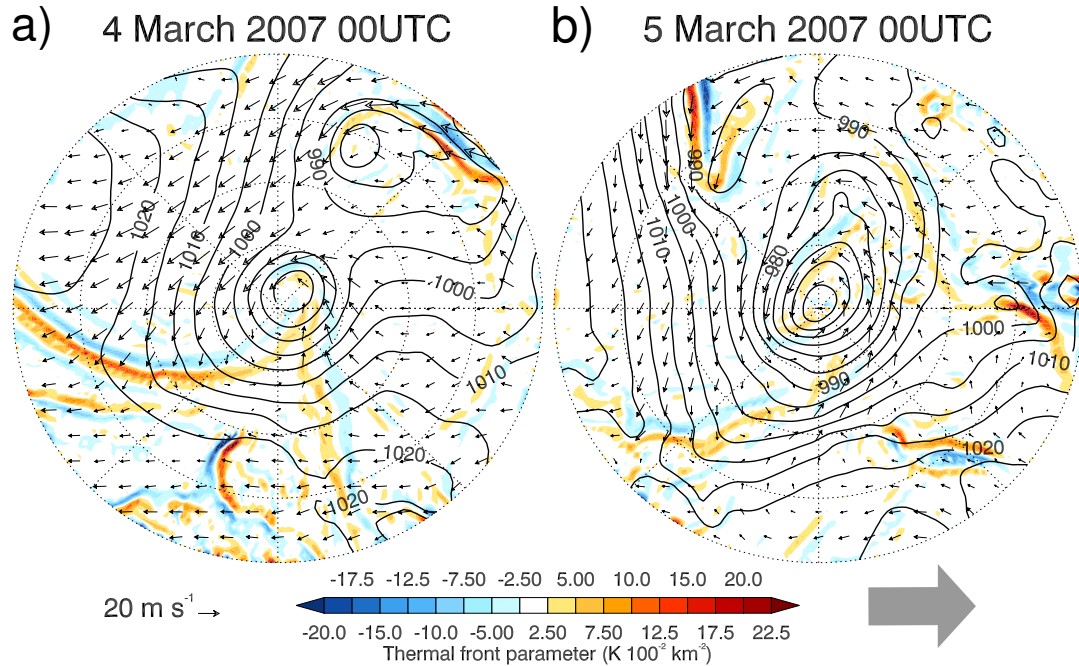

**Figure 5.** N1 cyclone-centered MSLP (solid black contours, 5 hPa contour intervals), 925 hPa system relative horizontal winds (20 m s$^{-1}$, reference arrow), and 925 hPa thermal front parameter (color, 2.5 K 100$^{-2}$ km$^{-2}$ intervals) on **(a)** 4 March 2007 00 UTC and **(b)** 5 March 2007 00 UTC. Radial dotted lines are plotted every 45° and dotted circles represent 5°, 10°, 15°, and 20° radii from cyclone center. The cyclone has been rotated in order that the direction of cyclone propagation is toward the right, indicated by the large grey arrow. N.b., geographical north for **(a)** and **(b)** is toward the right of the page.

levels of O$_3$ ($>$ 100 ppbv) above the dynamical tropopause (thick solid black contour), reaches down to nearly 500 hPa behind the cold front (-2° to 3°, Fig. 6b). There is further isentropic descent (along constant $\theta_e$ contours) of the relatively high levels of O$_3$ behind the cold front over a large area between -5° to 15°, with strong descent of $\omega \leq$ -30 hPa h$^{-1}$. This results in high O$_3$ ($>$ 55 ppbv) at 1000 hPa behind the surface cold front (-5° to 17°, Fig. 6b and c). The stratospheric fraction of air behind
5  the cold front is $\geq$ 30 % down to 700 hPa and $\geq$ 10 % down to 850 hPa (Fig. S2b).

A day later, the cold front associated with the N1 cyclone has passed over Mace Head and high O$_3$ (55 ppbv) is reported at the surface. There are elevated levels of O$_3$ in both reanalyses: the MACC reanalysis at the 1000 hPa near Mace Head $>$ 60 ppbv (Fig. 7c) and the MERRA-2 O$_3$ at 950 hPa is 45–50 ppbv (Fig. S3c). Since the cyclone has started to decay, it is more difficult to identify the fronts and airstreams within the cyclone (Fig. 5b). The maximum WCB ascent ahead of the
10  N1 cyclone at 500 hPa ($\omega >$ 30 hPa h$^{-1}$, Fig. 7a) is approximately half the value compared to the day before ($\omega >$ 60 hPa h$^{-1}$, Fig. 6a). However, there are strong values of positive $\omega$ at 500 hPa (Fig. 7a,d) associated with the upstream cyclones (closest cyclone is identified as a closed low contour to the northwest of the N1 cyclone in Fig. 7c). The DI airstream has also weakened, yet the tropopause is still visibly depressed at 400 and 500 hPa (Fig. 7b and d, respectively) and high levels of O$_3$

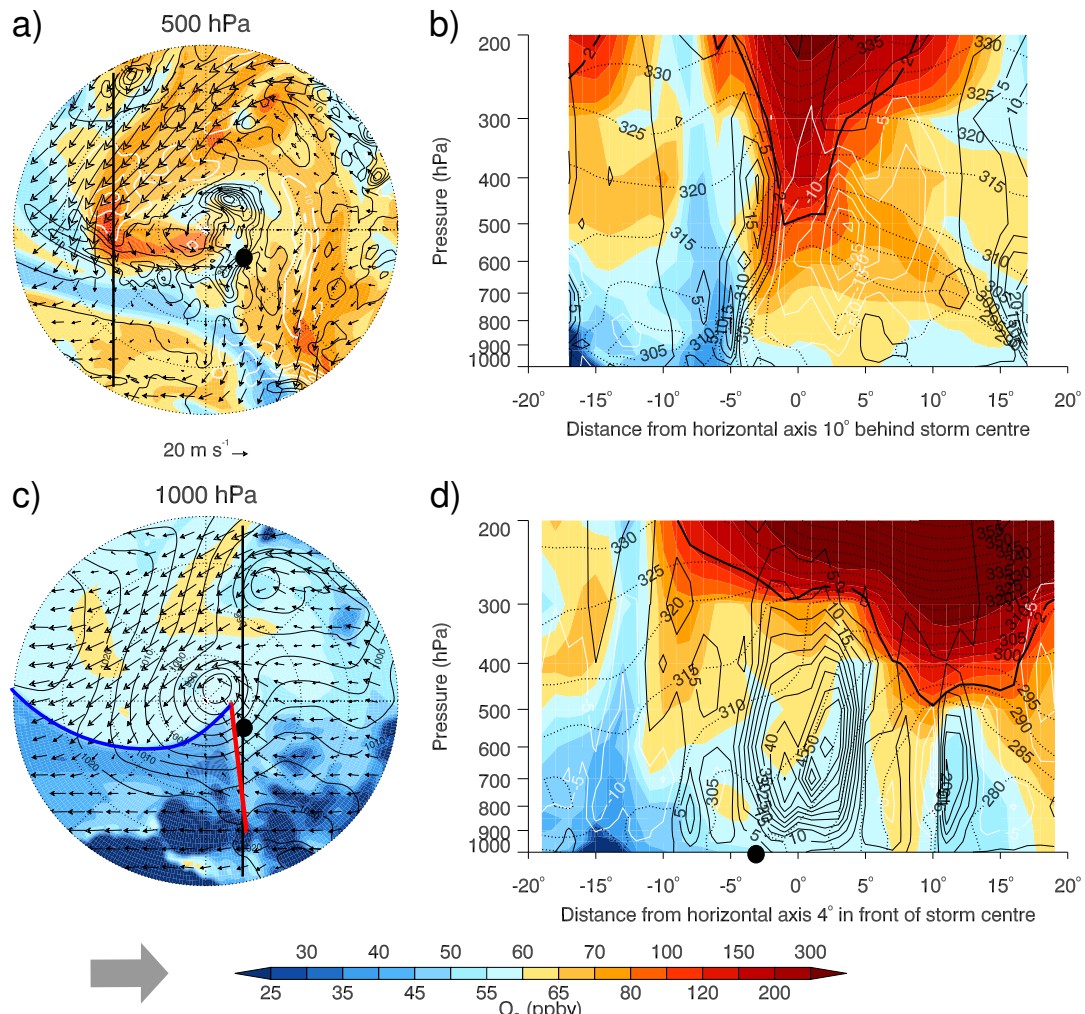

**Figure 6.** N1 cyclone-centered $O_3$ (color) and system relative winds (20 m s$^{-1}$, reference arrow) on 4 March 2007 00 UTC: **(a)** 500 hPa with $\omega$ (10 hPa h$^{-1}$ contour intervals, with black contours for positive values indicating ascent and white contours for negative values indicating descent) and **(c)** 1000 hPa with MSLP (solid black contours, 5 hPa contour intervals) and approximate location of the warm (red line) and cold (blue line) fronts based on Figs. 3b and 5a. The cyclone has been rotated in order that the direction of cyclone propagation is toward the right, indicated by grey arrow. The additional thick vertical lines in **(a)** and **(c)** indicate the location of the vertical transects (**b** and **d**) of $O_3$ (color), $\omega$ (5 hPa h$^{-1}$ contour intervals, with black contours for positive values indicating ascent and white contours for negative values indicating descent), $\theta_e$ (dotted contour lines, 5 K intervals), and the 2 PVU isosurface for the dynamical tropopause (thick solid contour): **(b)** 10° behind the cyclone center; and **(d)** 4° ahead of cyclone center, respectively. The cross sections are <40° diameter with positive axis values to the north. The approximate location of Mace Head is indicated by the large black dot (**a**, **c**, **d**).

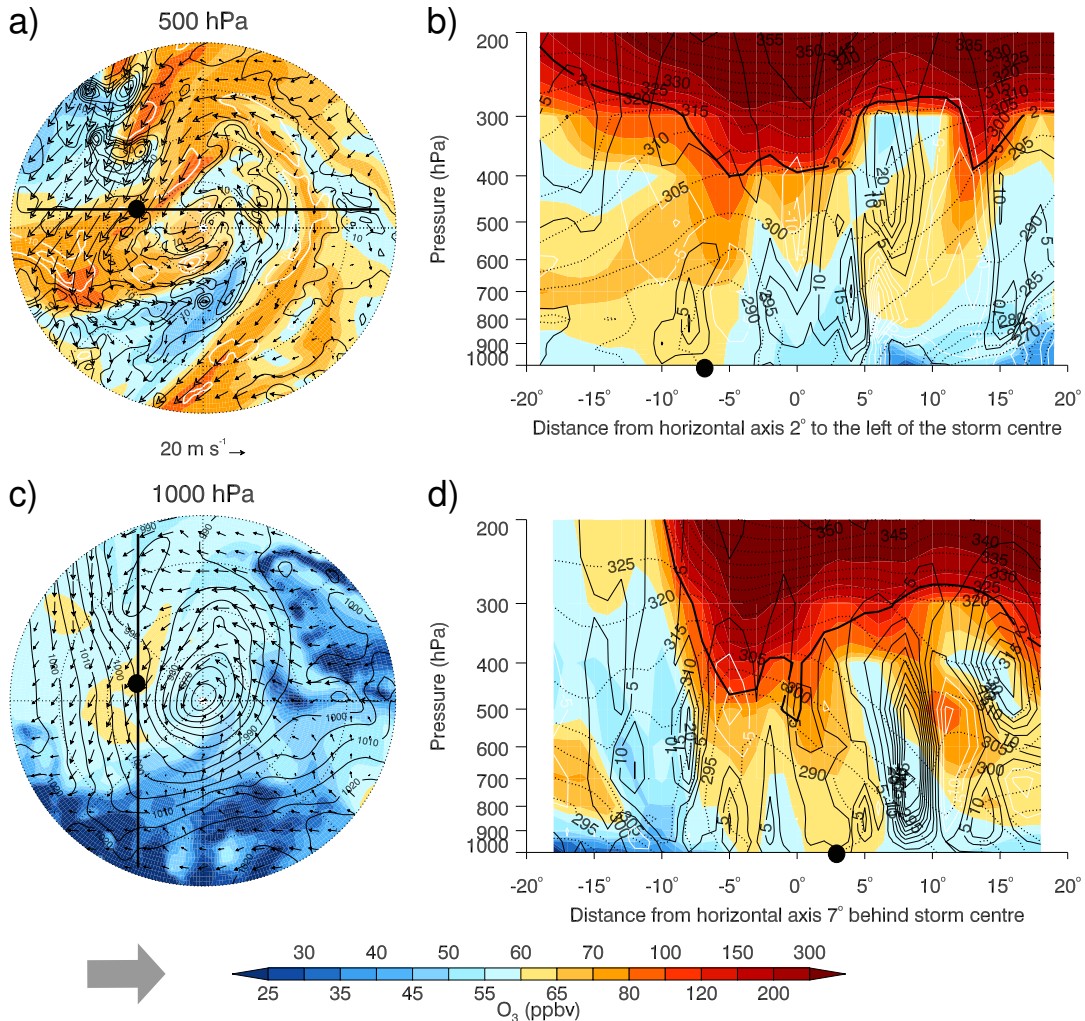

**Figure 7.** N1 cyclone on 5 March 2007 00 UTC, similar to Fig. 6, except 24 hours later. The additional thick vertical lines (**a**, **c**) indicate the location of the vertical transects (**b**) 2° to the north of the cyclone center and (**d**) 7° behind the cyclone center, respectively. The cross sections have positive axis values to the east (**a**, **b**) and to the north (**c**, **d**). The approximate location of Mace Head is indicated by the large black dot. The cyclone has been rotated in order that the direction of cyclone propagation is toward the right, indicated by grey arrow.

($> 55$ ppbv) are present throughout the mid- to lower troposphere (Fig. 7b,d). The $O_3$ originally from the stratosphere has descended isentropically toward the surface near the location of Mace Head (black dot, near -7° in Fig. 7b and 3° in Fig. 7d). This is also portrayed in the MERRA-2 $O_3$ in Fig. S3b,d, with a tongue of large STFR over Mace Head (at 700 hPa, STFR $> 40$ % and $> 30$ %, Fig. S4b and d, respectively). The high $O_3$ associated with the upstream cyclone descends into the mid-troposphere (10°, Fig. 7d) and is associated with high $O_3$ at Mace Head later in the life cycle of the N1 cyclone.

The MACC and MERRA-2 representations of the N1 storm (Figs. 6–7 and Figs. S1–S4, respectively), including the stratospheric $O_3$ mixing ratios, are similar. In the troposphere, MERRA-2 $O_3$ is expected to be biased low (Sect. 2.2.3), and while the MERRA-2 troposphere $O_3$ mixing ratios are lower in the N1 cyclone compared to the MACC reanalysis, there is still relatively higher $O_3$ within the DI airstream reaching the surface (Figs. S1b,d and S3b-d) as seen in Figs. 6 and 7. The higher vertical resolution of MERRA-2 shows the tropopause fold reaches just below 600 hPa (-3°, Fig. S1b) which is 100 hPa closer to the surface than shown in the MACC reanalysis (Fig. 6b). Since both reanalyses show similar $O_3$ spatial patterns, although not quantitatively the same, we are confident the $O_3$ reaching Mace Head has stratospheric origin.

Since the N1 cyclone was described in great detail, in the following case studies we will highlight differences compared to the N1 cyclone.

### 4.2.2 S1 cyclone: synoptic conditions associated with high $O_3$ at Mace Head

The S1 cyclone originated over New York, USA (43° N, 74° W) on 22 April 2012 and tracked past Mace Head through the South region between 24 April and 26 April 2012. High $75^{th}$ pc 6-hourly averaged $O_3$ ($> 45$ ppbv) was generally observed at Mace Head while the S1 cyclone was in the South region on 24 April at 12 UTC to 26 April 2012 at 06 UTC (Fig. 2a). The center of the S1 cyclone at the time of maximum vorticity (25 April 2012 at 00 UTC), is located almost directly to the south of Mace Head (50° N, 10° W, Figs. 2a and 8b) having merged with a downstream decaying cyclone ("P", Fig. 8b). At this time, Mace Head reported median $O_3$ levels. There is a high pressure system located directly to the west, centered at approximately 50° N, 45° W, which appears to block transport from upwind North American sources (Figs. 8b and 9d). The atmospheric circulation pattern of high pressure to the west and the S1 cyclone south of Mace Head is very different to the pattern seen during the life cycle of the N1 cyclone, where there were several cyclones in the NA sector and strong westerly winds due to the pressure gradient between the cyclones and the Azores High (Figs. 3 and 4).

The source of elevated surface $O_3$ ($> 55$ ppbv) at Mace Head is not associated with descent of air directly from the lower stratosphere within the DI airstream of the S1 cyclone, as seen in the N1 cyclone. Instead there is "residual" high $O_3$ in the mid- to lower troposphere (50°–60° N, 20° W–30° E, Fig. 8a). This $O_3$ is referred to as residual $O_3$ as it descended several days prior to the S1 cyclone passing near Mace Head (not shown) within the DI of the downstream cyclone ("P", Fig. 8b) prior to its decay. This high $O_3$ appears to become entrained on the north side of the S1 cyclone and advected cyclonically in the lower and mid-troposphere over northern UK and Eastern Europe/Scandinavia (Fig. 8).

Eighteen hours after the S1 cyclone reached maximum vorticity (25 April 2012 at 18 UTC), high levels of $O_3$ were reported at Mace Head. The MSLP field is still characterized by a blocking high in the western NA and the S1 cyclone is to the southeast of Ireland (Fig. 9d,h). Above Mace Head, there is descent (closed white contour, Fig. 9c,g) in the region of high $O_3$ ($> 55$ ppbv)

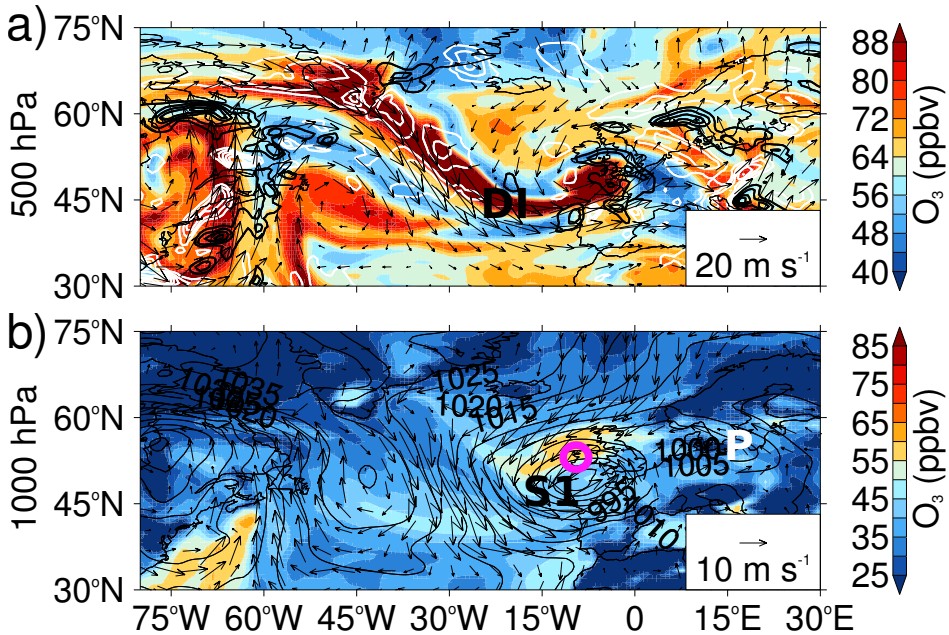

**Figure 8.** Synoptic conditions and $O_3$ distribution for the S1 cyclone (**b**; 12° N, 48° W) on the 25 April 2012 at 00 UTC, at the time of maximum $\zeta_{850}$. Two levels are shown: (**a**) 500 hPa and (**b**) 1000 hPa. Additional features are labelled including the DI airstream (**a**), the downstream decaying parent low to the S1 cyclone (white "P", **b**) and Mace Head is indicated (**b**).

which can facilitate the transport of $O_3$ from the mid-troposphere toward the surface. A large ridge has formed over the NA, identified at 300 hPa by the anticyclonic wind flow aloft as well as the 2 PVU contour (Fig. 9a). On either side of the ridge at 300 hPa, the troughs bring stratospheric air towards lower latitudes than Mace Head (Fig. 9a). This allows for $O_3$-rich air to be transported towards the surface at lower latitudes as part of the DI airstream within such strong cyclones as the S1 cyclone

5 (between 35° and 45° N, Fig. 9b-d).

To investigate the residual $O_3$ in more detail, we compare the S1 cyclone airstreams' vertical and horizontal distribution of $O_3$ at the time of maximum $\zeta_{850}$ (25 April 2012 00 UTC) using the cyclone-centered coordinate system in both the MACC and MERRA-2 reanalyses (Figs. 10 and 11). In Figs. 10a and 11a, the stratospheric $O_3$ within the DI airstream arrives into the center of the cyclone from the northwest quadrant and is bounded by the WCB ascent to the right of the cyclone's center

10 (black contours with maximum at 5° radius). Most of the northeast quadrant of the S1 cyclone at 500 hPa, shown in Figs. 10a and 11a, is dominated by the residual high levels of $O_3$ (> 55 ppbv; STFR is > 20 %, Fig. S6a) identified above as coming from the downstream decaying cyclone. The MACC and MERRA-2 $O_3$ mixing ratios throughout the mid-troposphere (above 600 hPa) to lower stratosphere (200 hPa) are comparable during the S1 cyclone (Figs. 10 and 11). The vertical transport of $O_3$ will be sensitive to the strong gradients of $O_3$ near the tropopause. The MACC $O_3$ within the DI airstream can be as much as

15 75 ppbv higher than the MERRA-2 $O_3$, as the MERRA-2 reanalysis $O_3$ values decrease closer to the dynamical tropopause

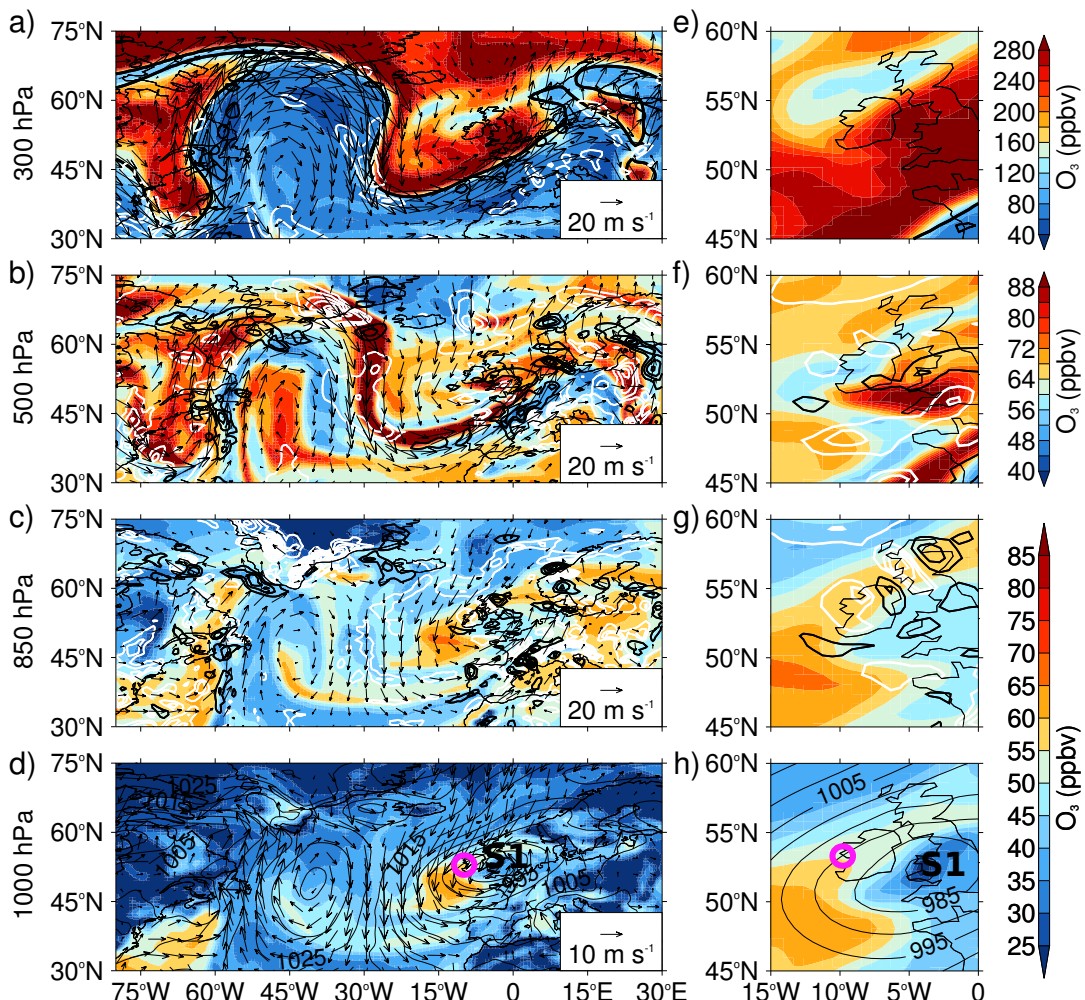

**Figure 9.** Similar to Fig. 8 (**a-d**) but also includes an inset over Ireland and the UK (45°–60° N, 15° W–0°; **e-h**, winds not included) for the S1 cyclone (**d**; 51° N, 5° W) 18 hours later, on the 25 April 2012 at 18 UTC for four pressure levels: (**a,e**) 300 hPa, (**b,f**) 500 hPa, (**c,g**) 850 hPa, and (**d,h**) 1000 hPa. High $O_3$ reported at Mace Head.

compared to the MACC $O_3$ distribution. Both the MACC and MERRA-2 $O_3$ have good agreement to independent observations in the upper troposphere/lower stratosphere (UTLS) region (Inness et al., 2013; Wargan et al., 2015, 2017). In the Northern Hemisphere (NH) extra-tropics, MACC $O_3$ between 500 and 100 hPa has a bias of -10–0% when compared to ozonesondes profiles (Inness et al., 2013) and Wargan et al. (2017) found that the average MERRA-2 $O_3$ profile over Europe (45°-60°N, 0°-60°E) in the spring of 2005 (after the assimilation of MLS and OMI satellite retrievals) captures the vertical structure of the ozonesondes in that region ($r2 > 0.6$), although biased low (up to -20 %) in the UTLS.

The two vertical cross-sections in Figs. 10b,d and 11b,d intersect the approximate location of Mace Head and the residual high $O_3$ in the mid-troposphere that is not associated with the S1 cyclone's DI airstream. The tropopause is depressed to about

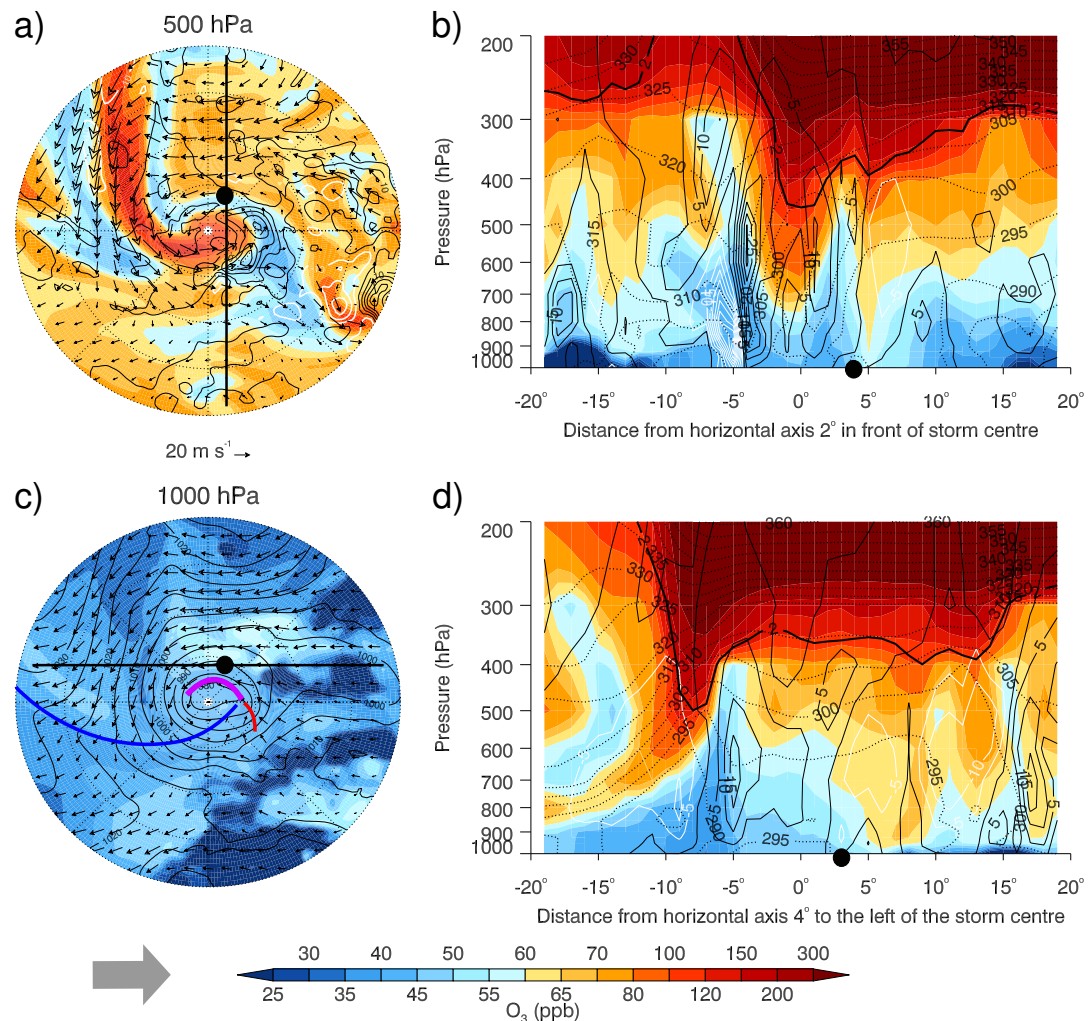

**Figure 10.** Similar to N1 cyclone Figs. 6 and 7 except for the S1 cyclone on 25 April 2012 00 UTC (see Fig. 8). The approximate location of the occluded (purple line), warm (red line) and cold (blue line) fronts in **c** are based on the fronts in Fig. S5a and the synoptic analysis chart (not shown). The additional thick vertical lines in (**a**) and (**c**) indicate the location of the vertical transects (**b**) 2° in front of the cyclone center and (**d**) 4° to the north of cyclone center, respectively. The cross sections have positive axis values to the north (**a**, **b**) and to the east (**c**, **d**). The approximate location of Mace Head has been indicated by large black dot. The cyclone has been rotated in order that the direction of cyclone propagation is toward the right, indicated by grey arrow.

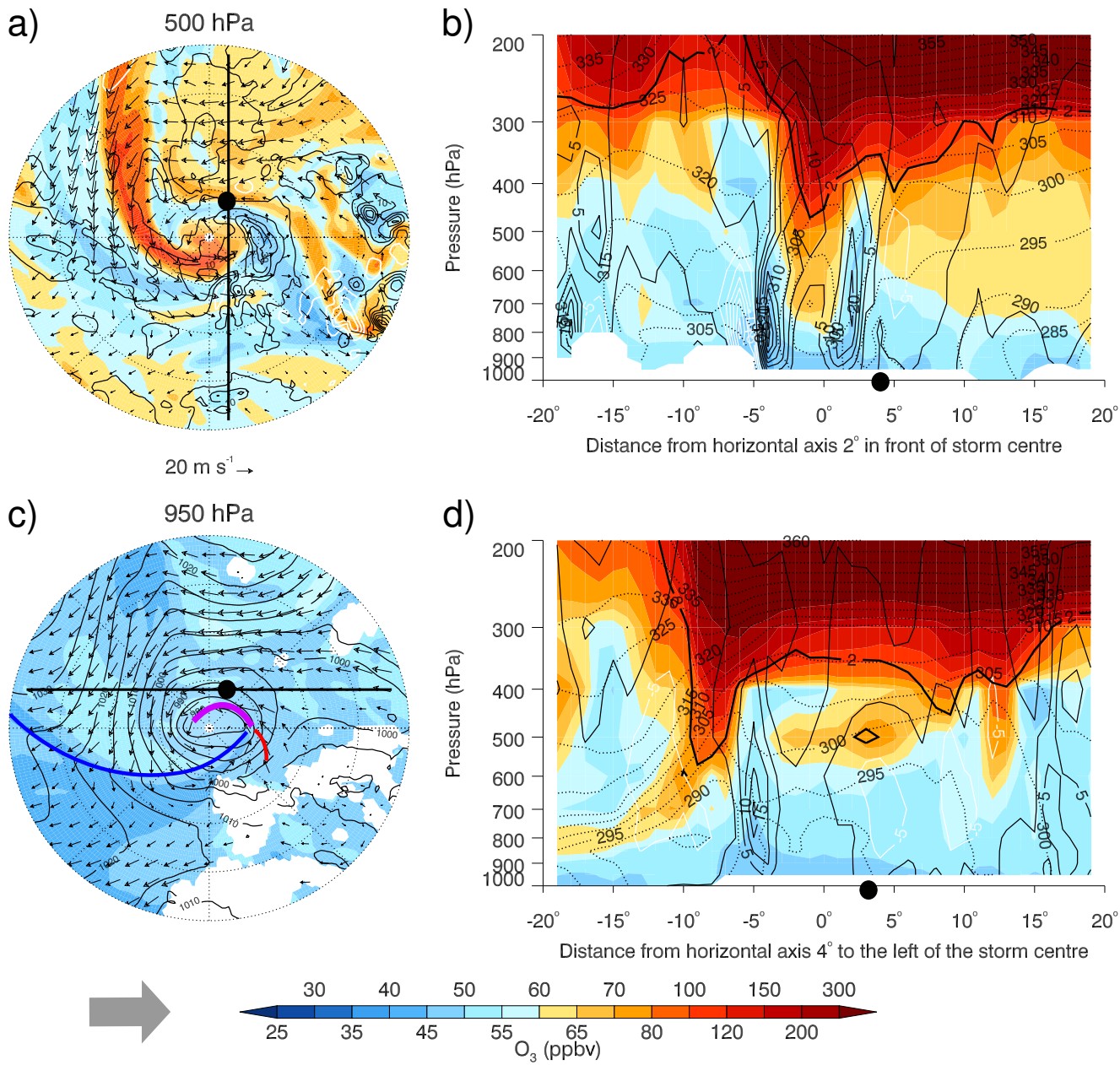

**Figure 11.** S1 cyclone on 25 April 2012 00 UTC, similar to Figs. 10 except using MERRA-2 reanalysis data. Note, panel **c** is 950 hPa instead of 1000 hPa to minimize the missing data plotted. This occurs since the GEOS-5 model masks orography.

500 hPa throughout the DI, shown close to the cyclone center (0°) in Figs. 10b and 11b and behind the S1 cyclone (-8°) in Fig. 10d and 11d. High levels of $O_3$ (> 55 ppbv) are found down to 800 hPa associated with the DI at the cyclone center (Fig. 10b and 11b) and within the isentropic descent behind S1 between -19° to -5° (Fig. 10d and 11d). There is no injection of stratospheric air into the troposphere occurring near Mace Head from the S1 cyclone, as the tropopause remains almost parallel
with the isentropes in the transects between 5° to 15° in Figs. 10b and 11b and between -4° to 15° in Figs. 10d and 11d. The MERRA-2 reanalysis shows the DI of the upstream parent low is directly related to the high $O_3$ in the mid-troposphere as a small closed 2 PVU contour circle – the dynamical tropopause – is above Mace Head at 500 hPa (Fig. 11d) which is not seen in the coarser MACC reanalysis (Fig. 10d). At 500 hPa, the STFR is > 40 % within the remnant $O_3$ associated with the decaying cyclone to the east of the S1 center (Fig. S6d). At about 5° in both Fig. 10b and d transects, there is descent (-5 hPa h$^{-1}$ contour)
of this residual high $O_3$ from the mid-troposphere toward the surface over Mace Head in the MACC reanalysis. Despite the negative vertical velocity above Mace Head in Fig. 11b and d, the simulated MERRA-2 $O_3$ does not descend fully to the surface, however the simulated $O_3$ at the surface at Mace Head in both reanalyses is approximately 45 ppbv.

### 4.2.3   N2 cyclone: synoptic conditions associated with high $O_3$ at Monte Velho

The N2 cyclone is a strong cyclone which passed north of Monte Velho and advected high $O_3$ to the observation site. Cy-
clogenesis of the N2 cyclone occurs off the east coast of Florida on 17 May 2006 at 12 UTC (not shown) and then tracks northeastward (Fig. 2b). The synoptic conditions at the time of the N2 cyclone are similar to the N1 cyclone: About a day after cyclogenesis, the N2 cyclone is located just south of Newfoundland and downwind of the N2 cyclone are a series of low pressure systems across the NA region as well as a strong Azores High stretching over most of the NA (Figs. 12a and S7c). The large pressure gradient results in strong westerly winds at near 45° N (Fig. 12a) throughout the troposphere (Fig. S7).
Similar to the N1 cyclone, stratospheric $O_3$ descends into the troposphere within the low pressure systems in the NA ahead of the N2 cyclone (Figs. 12a and S7). Similar to the S1 cyclone, a high pressure system develops to the west of the N2 cyclone (Fig. 12b-c), supporting upper-level descent behind the N2 cyclone's cold front and negating transport across the NA from North America (Figs. 12b-c, S8 and S9).

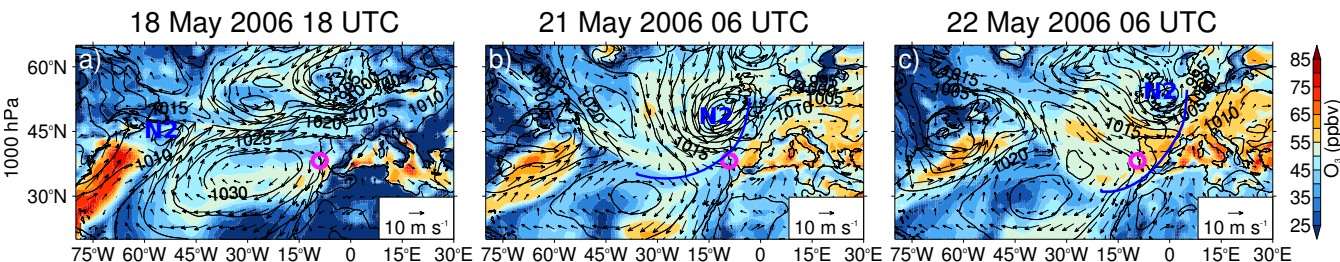

**Figure 12.** Synoptic conditions and $O_3$ distribution for the N2 cyclone at 1000 hPa: MACC $O_3$ (color) and horizontal wind vectors (10 m s$^{-1}$ reference arrow) and MSLP (solid contours, 5 hPa intervals) on **a**) 18 May 2006 18 UTC, **b**) 21 May 2006 06 UTC, and **c**) 22 May 2006 06 UTC. The approximate cold front (**b,c**; blue line) and Monte Velho (pink open circle) are included.

As was seen for the N1 cyclone, the N2 cyclone is also associated with low $O_3$ before high $O_3$ is observed at the monitoring station due to the passage of a cold front (Fig. 12b-c). Since the N2 cyclone is similar to the N1 and S1 cyclones discussed in Sects. 4.2.1 and 4.2.2, full details for this case can be found in the Supplemental material (Sect. S2.3).

### 4.2.4  S2 cyclone: synoptic conditions associated with high $O_3$ at Monte Velho

The S2 cyclone is a moderate cyclone that formed on 12 March 2007 00 UTC to the south of a large high pressure system which extended over the eastern NA and Europe (not shown). On 13 March 2007 18 UTC, 42 hours later, the S2 cyclone reaches maximum vorticity ($\zeta_{850}$ =7.3x10$^{-5}$ s$^{-1}$, 31.3° N, 16.5° W) and is still located to the south of the high pressure system over the eastern NA (Fig. 13d). At this time, there are additional strong weather systems in the NA region: a deep Icelandic low, a low over Hudson Bay and a high pressure system over the western NA. Within the S2 cyclone, there is strong descent within the DI airstream between 300 and 850 hPa (Fig. 13a-c), with maximum descent at 500 hPa ($\omega <$ -25 hPa h$^{-1}$, Fig. 13b) nearly as strong as the N1 cyclone (Fig. 6b). High values of $O_3$ ($>$ 80 ppbv) descend from upper levels toward the surface within the DI airstream and advect cyclonically around the S2 cyclone and anticyclonically within the high pressure system at lower levels, behind the low-level cold front (850 and 1000 hPa, Fig. 13c and d). Over the Iberian Peninsula at 850 hPa, there is descent of $O_3$-rich air likely associated with the large region of descent throughout the troposphere over the Mediterranean Sea associated with a downstream cyclone (Fig. 13a-c).

Over the following four days, the high $O_3$ at Monte Velho associated with the S2 cyclone persists while the S2 cyclone tracks southeastward (see Fig. 2b). At upper-levels, a closed low detaches from the stratosphere (not shown). In Fig. 14a, the descent within the DI of the S2 cyclone can be identified at 500 hPa, however the S2 cyclone can no longer be identified by a closed low pressure system over Northern Africa in Fig. 14c. The high pressure system over the eastern NA moves southwestward with the $O_3$-rich air which has accumulated within it (Fig. 14c). There is still descent occurring over the Iberian Peninsula in the mid-troposphere, which continues to transport relatively high levels of $O_3$ to the surface at Monte Velho (Fig. 14a and b).

The cyclone-centered analysis was not performed for the S2 cyclone since 1) the high $O_3$ at Monte Velho prior to maximum vorticity is associated with a downstream cyclone's DI airstream (as seen in the S1 cyclone) and 2) after the S2 cyclone started to decay the high $O_3$ is associated with subsidence in the high pressure system behind the S2 cold front.

## 5   Discussion

The influence of springtime mid-latitude NA cyclones on $O_3$ observations at two marine background sites – Mace Head, Ireland and Monte Velho, Portugal – has been quantified using a frequency analysis test and explored through cyclone case studies. About 50 % of the cyclones which track to the north or close to Mace Head show a relationship with high levels of $O_3$ ($> 75^{th}$ pc) observed at Mace Head, while 53 % of the tracks to the south of Mace Head are associated with low levels of observed $O_3$ ($< 25^{th}$ pc). The N1 case study cyclone to the north of Mace Head illustrates how $O_3$-rich air, which originated in the stratosphere, reaches the surface within the DI airstream as the cyclone passes Mace Head. In contrast, at the southern measurement station, Monte Velho, the levels of $O_3$ are not strongly influenced by the passing cyclones to the north or to

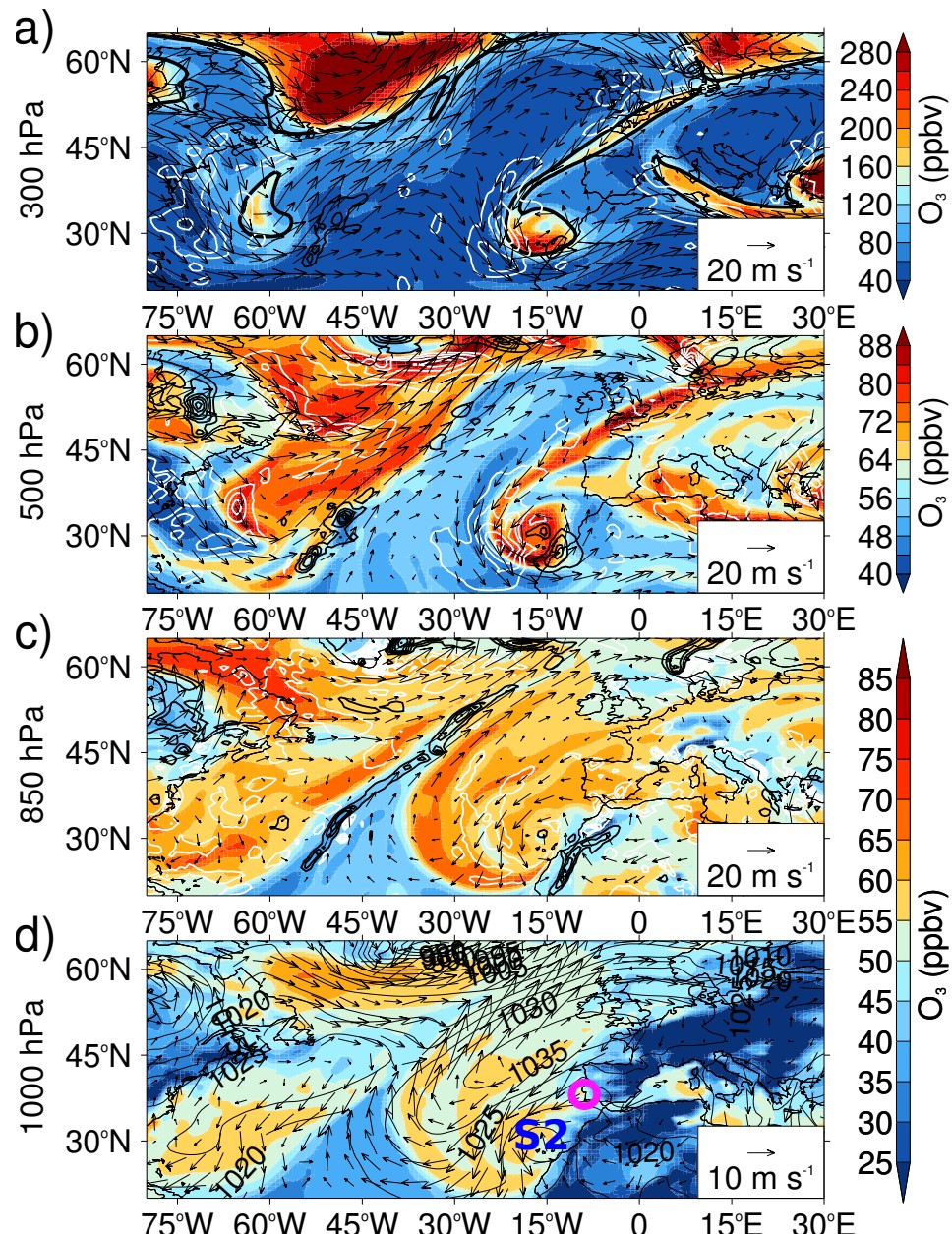

**Figure 13.** Synoptic conditions for the S2 cyclone (**d**; 31° N, 17° W) on the 13 March 2007 18 UTC, at the time of maximum $\zeta_{850}$. $O_3$ (color; note different scales used for different levels) and horizontal winds (note magnitude of reference arrow changes) are shown on four levels: (**a**) 300 hPa, (**b**) 500 hPa, (**c**) 850 hPa, and (**d**) 1000 hPa. In addition, MSLP (**d**; solid contours, 5 hPa intervals), $\omega$ (**a**, **b**, and **c**; 5 hPa h$^{-1}$ contour intervals, black contours for positive values indicating ascent (n.b., different interval to previous cyclones) and white contours for descent), and 2 PVU isosurface (**a**; thick contour) are shown. Monte Velho indicated by pink open circle.

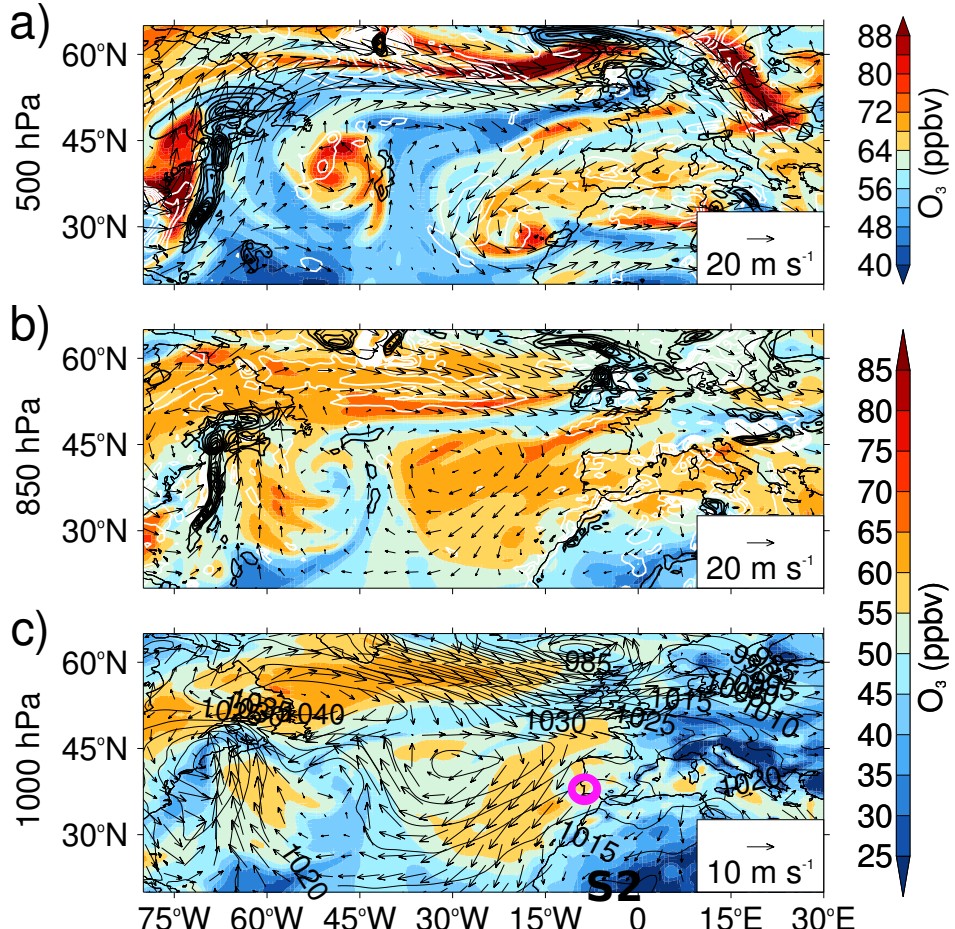

**Figure 14.** Similar to Fig. 13 except for the S2 cyclone (**c**; 23° N, 1° W) nearly 4 days later on the 17 March 2007 12 UTC. High $O_3$ recorded at Monte Velho. $O_3$ (color; note different scales used for different levels) and horizontal winds (note magnitude of reference arrow changes) is shown on three levels: **(a)** 500 hPa, **(b)** 850 hPa, and **(c)** 1000 hPa. In addition, MSLP (**c**; solid contours, 5 hPa intervals) and $\omega$ (**a** and **b**; 5 hPa h$^{-1}$ contour intervals, black contours for positive values indicating ascent and white contours for descent) are shown.

the south. If the tracks pass close to Monte Velho, 55 % of tracks are associated with high levels of $O_3$. The location of the Monte Velho site relative to the Azores High is the determining meteorological factor for the temporal variability of $O_3$ at this site since Monte Velho is located south of the NA storm track. Within the Azores High, neighboring the cyclone, there is build-up of high concentrations of $O_3$ related to the anticyclonic descent within the DI airstream behind the cold front.

5      In this study, up to 30 % of springtime cyclones are associated with both high $O_3$ and low $O_3$ observations at Mace Head or Monte Velho associated with frontal passage relative to the measurement sites. For example, the difference in modelled MACC $O_3$ across the N1 cyclone's cold front is > 10 ppbv, with high $O_3$ behind the cold front and low $O_3$ ahead of the cold front. Prior to the N1 cyclone reaching maximum intensity, Mace Head is located in the warm sector of the cyclone with moderate

levels of $O_3$ (< 45 ppbv) arriving from the southwest possibly south of 30° N. Previous studies have associated air arriving at Mace Head from the NA with relatively clean maritime air masses (e.g. Jennings et al., 1991; Simmonds and Derwent, 1991; Simmonds et al., 1997; Derwent et al., 1998). Simmonds et al. (1997) noted $O_3$-poor air, around -5 ppbv relative to the background levels, arrives at Mace Head in April and May from the southwest with subtropical signatures (Simmonds et al., 1997).

It is shown here that high levels of $O_3$ can be associated with the natural sources of $O_3$ transported from the stratosphere and can arrive at Mace Head from the west. The stratospheric $O_3$-rich air descends toward the lower levels within the DI airstream following direct STT, such as in the N1 intense cyclone and the S2 moderate cyclone. However, the S1 and N2 strong cyclones show that $O_3$-rich air in the mid-troposphere can be older, undiluted ("residual") $O_3$-rich air due to STT occurring in a previous downstream cyclone which then becomes entrained in the new cyclone and descends. In the S1 cyclone near Mace Head, $O_3$-rich air from the DI of a downwind cyclone is found in the mid-troposphere (700 to 400 hPa) and is advected cyclonically around the cyclone subsequently descending toward the surface in the DI airstream. As a result, high levels of $O_3$ (> 55 ppbv) at Mace Head persists for about a day and a half. This feature of older $O_3$-rich air descending toward the surface in the DI airstream in combination with subsidence in the anticyclone behind the cyclone was found in the N2 and S2 case studies. Cooper et al. (2004a) also found that $O_3$-rich air that had descended in the DI of single cyclone was advected westward within a high pressure system behind the cyclone and then became entrained within an upstream cyclone's WCB. A number of studies suggest it can take many days for $O_3$-rich air to be diluted into the free troposphere, in which time there are several transport pathways before the air eventually reaches the surface (Johnson and Viezee, 1981; Colbeck and Harrison, 1985; Davies and Schuepbach, 1994; Cooper et al., 2004a). In the case study cyclones examined here, the main pathways are direct entrainment into the cyclone or an upstream cyclone and subsidence within a neighboring anticyclone.

In all four case studies, stratospheric intrusions do occur within the DI airstream regardless of the cyclone intensity. In particular, the moderate S2 cyclone, to the south of Monte Velho, exhibits strong descent from upper levels ($\omega$ < -25 hPa h$^{-1}$) comparable to the N1 cyclone and greater than the descent within the N2 and S1 cyclones. Based on stratospheric intrusions targeted by 10 flights over Central USA in the spring and fall of 1978, Johnson and Viezee (1981) conclude that all low pressure troughs may be associated with STT, however the amount would be proportional to the strength of the trough. Cooper et al. (1998) used ozonesondes to sample pre- and post-frontal vertical profiles at Bermuda and two US sites and found that the mean tropospheric $O_3$ always increased with height after the cold front had passed, with back trajectories indicating the source was the UTLS to the northwest. The stratospheric "influence" tracer from Ott et al. (2016) was used to represent the fraction of stratospheric air within the case study cyclones; however, in order to diagnose the amount of $O_3$ from the stratosphere that is irreversibly transferred into the troposphere a chemistry transport model or a chemistry climate model – such as GEOS-5 Chemistry-Climate model – should be run in order to evaluate the fluxes of $O_3$ or tagged-tracers of $O_3$, neither of which are archived in the MACC or MERRA-2 reanalyses. In addition, other sources of both natural and man-made pollutants could become entrained into the DI airstream (e.g., Brown-Steiner and Hess, 2011; Lin et al., 2012b; Langford et al., 2017). Biomass burning plumes from as far away as Canada have reached Europe, leading to enhanced CO at Mace Head as the plume passes

over (Forster et al., 2001). Within a fire plume, $O_3$ can be produced associated with increased precursors and this has the potential to enhance European $O_3$, including at Mace Head. This work is ongoing.

The case study analysis focused on the strongest cyclones associated with high $O_3$ at Mace Head and Monte Velho such that 1) the cyclone tracks would be exact matches in the MACC and ERA-Interim reanalyses (Knowland et al., 2015) and 2) airstreams would be most evident (Catto et al., 2010). An alternative method to select case study cyclones would be to start by selecting the tracks associated with the highest (or lowest) $O_3$. This leads to the selection of moderate to weak cyclones (which are generally not located solely in the major storm track pathways and which tend to be shorter-lived (Knowland et al., 2015)). While the case study analysis presented here highlights the regional differences in the strongest springtime mid-latitude cyclones associated with high surface $O_3$, the relationship between cyclones and surface $O_3$ likely depends on a number of complex factors, including the cyclone location relative to the measurement station and upstream and downstream pressure systems, cyclone intensity, stratosphere-to-troposphere transport, and long-range transport of trace gases within the free troposphere and the boundary layer.

For the case studies, it is difficult to identify anthropogenic $O_3$ from surface emission sources arriving at Mace Head or Monte Velho from the west across the NA since anthropogenic $O_3$ can be an order of magnitude smaller than the levels of stratospheric $O_3$ from STT. Regardless, Li et al. (2002) show North American major transatlantic pollution events are infrequent, identifying only 2 such events during the spring of 1997. For pollution from North American emission sources to reach western Europe, the perfect set up of sequential low pressure systems and a strong Azores-Bermuda high, as seen during the life cycles of the N1 and S2 cyclones, are required to rapidly transport the pollutants in the resultant strong westerly winds (Li et al., 2002; Creilson et al., 2003). Li et al. (2002) found in their study that such events are more likely to occur during positive phases of the North Atlantic Oscillation (NAO; Walker and Bliss, 1932). The month of March 2007 which included the N1 and S2 cyclones was characterized by a positive NAO index (1.44, Climate Prediction Center (Last retrieved 19 September 2014)). The synoptic set-up during the S1 and N2 cyclones had a blocking high directly to the west, representative of the negative NAO phase; the NAO index was weakly positive (0.47) and negative (-1.28) during the months of April 2012 and March 2006, respectively. The relationship between the NAO, STT and pollution transport has been explored previously (e.g. Creilson et al., 2003; Sprenger and Wernli, 2003; Pausata et al., 2012), however the direct impact of the cyclones and their associated airstreams on the distributions of trace gases such as $O_3$ during the different NAO phases will be addressed in a further study.

This analysis has built upon the findings of earlier studies of airstream analysis and chemical composition (e.g. Bethan et al., 1998; Cooper et al., 2001, 2002a, b; Owen et al., 2006; Knowland et al., 2015) to identify consistent pathways in which relatively high levels of $O_3$ are transported through cyclones in the NA during spring. From a composite of intense springtime cyclones which included the N1 cyclone, Knowland et al. (2015) characterized the main mechanisms which advect $O_3$ horizontally and vertically within cyclones and hence the extent to which the airstreams can influence $O_3$ concentrations throughout the troposphere. The case studies corroborate with the findings of the composite study that a typical NA cyclone WCB is often associated with relatively lower levels of $O_3$ while the DI is associated with relatively higher levels of $O_3$ connecting the stratospheric reservoir with the surface (a Type 3 intrusion as defined by Johnson and Viezee (1981)). In

addition, the case studies depict elevated $O_3$ from either a parent low pressure system (e.g., the N1 cyclone case study) or an upstream decaying cyclone (e.g., the S1 and N2 cyclones) which can become entrained within the cyclone; this is a strong feature in the NA cyclones as there is elevated $O_3$ (up to 5 ppbv greater than the background composite) similarly to the north of the NA composite cyclone center, not within the main DI airstream and separated from the $O_3$-poor WCB, that wraps cyclonically around the composite cyclone (Knowland et al., 2015).

## 6 Conclusions

We have shown that passing cyclones have a discernible influence on surface $O_3$ concentrations; especially when cyclones track north of 53° N, there is a significant probability that the surface $O_3$ at Mace Head will be high ($> 75^{th}$ pc). Cyclones are more likely to be associated with both low $O_3$ ($< 25^{th}$ pc) and high $O_3$ if the cyclones track to the south of the observation site or if the cyclones are to the west of the observation site.

The four case studies demonstrate the main drivers associated with cyclones that impact surface $O_3$ concentrations: a) the importance of the passage of a cyclone's cold front to influence surface $O_3$ measurements, b) the ability of cyclones to bring down high levels of $O_3$ from the stratosphere and c) that accompanying surface high pressure systems play an important role in the temporal variability of surface $O_3$. From these four case studies we have seen that the location of the different fronts at different stages in the cyclone life cycle influence the surface $O_3$ reported at these two observation sites. Specifically:

- Prior to the passing of the N1 and N2 cold fronts, low-level, $O_3$-poor subtropical air within the WCB arrived from the southwest to Mace Head and Monte Velho, respectively. After the cold fronts had passed over the respective observation site, high surface $O_3$ was reported.

- The N1 cyclone showed a direct connection with stratospheric $O_3$-rich air descending within the DI airstream which reaches Mace Head after the cold front passes. Within the S1 and N2 cyclones, there is aged stratospheric $O_3$-rich air entrained from previous downwind cyclones. This $O_3$-rich air subsequently descends within the strong cyclone and impacts the surface $O_3$ observations after the cold front has passed over the site. Strong descent of stratospheric $O_3$ within the DI airstream occurs within moderately strong cyclones outside the main storm track as demonstrated during the lifetime of the moderately strong S2 cyclone to the south of Monte Velho.

- During the life cycles of the S1 and N2 cyclones, there are blocking high pressure systems in the western NA which cut off any transport of $O_3$-rich air from upstream sources. An important descent mechanism was identified during the life cycle of the S2 cyclone, whereby subsidence and accumulation of $O_3$-rich air within the neighboring Azores High pressure system impacted $O_3$ measurements at Monte Velho while the S2 cyclone tracked over northwest Africa.

This study has shown that there are diverse cyclone-related dynamical processes which influence the variability of surface $O_3$ levels at two remote European coastal measurement sites on short time scales of hours to days. In addition, the spatial variability in the low and high pressure systems over the North Atlantic can distinctly influence the $O_3$ values observed at near sea level.

*Data availability.* This study used publicly available O$_3$ observations supported by EMEP (http://actris.nilu.no). The ERA-Interim reanalysis and the MACC reanalysis are available for download from ECMWF (http://apps.ecmwf.int/datasets/). The MERRA-2 reanalysis data is available through the NASA GES DISC online archive (https://disc.gsfc.nasa.gov/uui/datasets). The GEOS-5 STFR data will be provided upon request by Lesley Ott.

5    *Competing interests.* There are no competing interests.

*Acknowledgements.* This research was funded in part by the University of Edinburgh through the Principal's Career Development Scholarship, the Edinburgh Global Research Scholarship Award, and by the School of Geosciences, University of Edinburgh. K. Emma Knowland's analysis of MERRA-2 was supported through GMAO core funding administered by NASA's Modeling, Analysis, and Prediction Program. Implementation of STFR in GEOS-5 was funded by NASA's Atmospheric Composition Campaign Data Analysis and Modeling program.
10    MERRA-2 data were provided by the GMAO at NASA GSFC through the NASA GES DISC online archive. The authors would like to thank Diamantino Henriques and Lourdes Bugalho from Instituto Português do Mar e da Atmosfera (IPMA) and Paulo Beliche from Comissão de Coordenação e Desenvolvimento Regional (CCDR) do Alentejo for going to great lengths to confirm the instruments used at the Monte Velho, Portugal monitoring station.

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
