# Peer review of "The influence of mid-latitude cyclones on European background surface ozone"

_Atmospheric Chemistry and Physics, 2017_

## Referee Comment (RC1) · Anonymous Referee #2 · 21 Jun 2017

Review of manuscript acp-2017-318

**The influence of mid-latitude cyclones on European background surface ozone,**

by K. E. Knowland et al.

May 19, 2017

**General Remarks**

The topic of an elevated stratospheric ozone contribution at the surface has been in the focus of numerous recent studies driven by air-quality considerations. In particular, it has has been a mystery to me why Mace Head is reached by descending air masses, which has not been observed that frequently elsewhere. The paper is well written and deserves publication, but I recommend considering a few amendments.

**Details**

(1) Introduction: Consider mentioning the role of STT in some of the recent air-quality studies, such as:

Lefohn et al., Atmos. Environ. 45 (2011), 4845-4857
Langford et al., JGR 117, D00V06 (2012)
M. Lin et al., JGR 117, D00V22 (2012)
Lefohn et al., Atmos. Environ. 62 (2012), 646-656
Ma et al., ACP 14 (2014), 5311-5325
Dempsey, Atmos. Environ. 98 (2014), 111-122
M. Lin et al., Nature Communications 2015
Itoh and Narazaki, ACP 16 (2016), 6241-6261
Langford et al., JGR 122 (2017), 1312-1337

High-lying sites are more affected by STT (western US, Alps (Zugspitze, Jungfraujoch), Tibet).

(2) Introduction/Discussion: The penetration of STT air into the PBL has been rarely observed (you cite Davis and Schuepbach (1994); some more words can be found in Eisele et al., J. Atmos. Sci. 56 (1999), 319-330). The probability of entrainment into the PBL seems to be low, which suggests subsidence during night-time followed by day-time turbulent mixing (Eisele et al., 1999; Ott et al., JGR 121 (2016), 3687-3706; Langford et al., 2017) to be a more reasonable mechanism. Is there anything different at the seaside (Cooper et al., JGR 100 (2005), D23310; Itoh and Narazaki (2016))? You mention a fraction around 50 %. Could this be related to the chance of night-time subsidence?

(3) Also constituents from originally lofted Canadian fire plumes were observed at Mace Head (Forster et al., JGR 106 (2001), 22887-22906

(4) P. 4, line 28: The spatial resolution seems to be sufficient to resolve STT layers. This is an important issue (e.g., Roelofs et al., J. Geophys. Res. **108** (2003), 8529; Eastham and Jacob, ACP 17 (2017), 2543-2553). However, there is, still, the issue of mixing even in high-resolution models that can lead to a loss of information (two rather bad comparisons between

measurements and a high-resolution ECMWF model results: Trickl et al., ACP 16 (2016), 8791-8815). This topic should be addressed somewhere in the paper.

---

## Referee Comment (RC2) · Anonymous Referee #1 · 24 Jun 2017

**Review of the manuscript „The influence of mid-latitude cyclones on European background surface ozone" by K.E. Knowland et al., 2017**

**acp-2017-318**

This study investigates the influence of springtime cyclones on surface ozone concentrations of two surface measurement sites in Ireland (Mace Head) and Portugal (Monte Velho). After a statistical evaluation of reanalysis data, the authors present four case studies - two for each measurement site – of cyclones leading to high ozone values at the surface. The results show that cyclones passing the measurement sites within a certain range can have a recognizable effect on the surface ozone concentration – either by transporting ozone-rich air directly from the stratosphere or by lower aged stratospheric air.
This manuscript is interesting and very carefully written and therefore in the scope of ACP.

**Major comments**

My only major comment deals with the cyclone intensity. Why did you chose the most intense cyclones? Would it be better to investigate in addition also "normal" or "weaker" cyclones to get a broader picture of the influence of cyclones on surface ozone? Do only strong cyclones have this impact on the ground stations? Is there a clear relationship of high ozone and intense cyclones? Can weaker cyclones have a stronger impact on the measured ozone by advecting aged low-level ozone-rich air to the measurement sites?

Why not selecting the case studies by using only the criteria regarding the high ozone values (two or more time steps with >75th pc $O_3$)? Maybe you can illustrate your motivation of the cyclone selection in more detail.

**Technical corrections**

Page 4, line 31: Do not use italics for units. Use $m^2\ kg^{-1}\ s^{-1}$instead of *$m^2\ kg^{-1}\ s^{-1}$*

Section 2.2.2 Please indicate already here the spatial resolution of the MACC data (it follows in Sec 3.1, but for completeness it would be nice to have it already in the data description part)

Page 9, line 14: Are two time steps enough to justify the term "persistent"?

Figure 4: something went completely wrong with the labelling (see screenshot) -> please check

[Figure]

Figure 11: a,b,c,d labels are missing.

---

## Referee Comment (RC3) · Anonymous Referee #3 · 3 Jul 2017

General comments:

This manuscript investigates how passing mid-latitude cyclones influence the surface ozone levels at two stations in Western Europe (Mace Head and Monte Velho) by combining ozone observations and reanalysis data from ERA-Interim, MACC, and MERRA-2. Four case studies were presented to support the main results. The strong impact of the downward transport of O3-rich air masses from the stratosphere on the surface ozone levels was pointed out.

The topic and the general length of the paper are relevant for publication in ACP. Generally the paper is compact and well-written, however Sect. 4.2.1-4.2.3 should be improved as suggested below. The structure of the paper is logical. The introductory discussion is adequate and the referencing is sufficient. In a few cases more details on

the methods are needed. Some of the figures need improvement.

For the reasons mentioned above and below the paper is appropriate for publication in ACP after a major revision.

Specific comments:

- Shorten Sect. 4.2.1 and improve the text of this section (write more compact and structured). Perhaps add a table to shorten the text.

- Section 4.2.2 should be better weighted (better incorporate the amount of text compared to the number of figures).

- In comparison Section 4.2.3 is extremely brief and all information is shifted to the Supplement. → I get the general impression that Sect. 4.2.1-4.2.3 must be better weighted and the text must be improved and more compact written. Sometimes hard to follow for the reader.

- Sometimes the numbers for statistics are rather low (number of strong cyclones, see minor comments). Comment on this!

- Perhaps it would also be useful to show observed ozone time series for the whole months of your case studies?

Your results must be very sensitive to the vertical distribution of ozone in the tropopause layer (reanalysis data). Have you investigated this in detail? How accurate are the reanalysis data at these altitudes?

Minor comments and technical corrections:

Page 2, line 7: Add also the mean latitude for the Islandic low, since you also give this for the Azores High (line 9).

Page 2: Add this reference to your discussion: Trickl, T., O. R. Cooper, H. Eisele, P. James, R. Mücke, and A. Stohl (2003), Intercontinental transport and its influence on

the ozone concentrations over central Europe: Three case studies, J. Geophys. Res., 108, 8530, doi:10.1029/2002JD002735, D12.

Page 3, line 25-30: What kind of instrument was used for the ozone measurements?

Page 3, line 29: Replace "pink" by "orange".

Page 4, line 5: More clearly point out the difference between the present paper and Knowland et al., 2015.

Page 4, line 12-13: Add webpages for ERA-Interim, MACC, and MERRA-2 reanalysis.

Page 4, line 25: What does "cycle 31r2" mean?

Page 5, line 2: What does "cycle 36r1" mean?

Page 5, line 13: What is the explanation for the underestimate?

Page 6, line 2-5: Reorder the references in chronological order (and for same years in alphabetical order).

Page 6, line 11: Write out STFR.

Page 6, line 19: Reorder the references in chronological order.

Page 6, line 27: "<1000 km" in what time?

Page 8, Figure 1: The letter size of lat and lon numbers is too large compared to the track density numbers.

Page 9, Table 1: The number of strong cyclones used in the study is very low. Is this number high enough for your statistics?

Page 10, line 23: What does "SGF" mean?

Page 11, Table 2: Last column (right): "percent" → "Percent"

Page 11, Table 2: The number of years given in the brackets (SGF) is very low. Can

you comment on this low number for your statistics?

Page 14, Figure 3 legend: Always mention latitude first and then longitude. Check this throughout the paper.

Page 15, Figure 4: It is very hard to recognize Ireland in the maps (especially in b and c).

Page 16, line 34: Not common to mention Fig. 6c before Fig. 6a.

Page 18, Figure 6: the letters "a)" and so on are too big in comparison to the numbers along the axes. Reduce this for all figures throughout the paper.

Page 18, Figure 6: In a) the white lines are very hard to see.

Page 19, line 27-29: This is a good example of the sentences that should be improved in this section (as mentioned in "Specific comments").

Page 21, line 22: White contours are hard to recognize.

Page 21-26: In this section you present many figures, however very little text to the figures. Weight figures and text better.

Page 31, line 15-16: This sentence is somehow stand-alone. What is the message?

Page 31, line 18: "We have shown passing cyclones have" → "We have shown that passing cyclones have"

Page 31, line 26: For a better structure after "Specifically:" Add a "-" to all small paragraphs that follow and use an insertion slide-in.

Supplement: Figs. S2c, S4c, and S6c are not very useful, since they are too dark.

———————————————

---

## Author Comment (AC1) · 6 Sep 2017

The authors would like to thank the referees for their constructive reviews. In response, we have added text to the Introduction and Discussion sections to address Referee #2's comments and clarifying our Methodology to address Referee #1's concerns. The Results section has been revised in order to better weight the text and figures as suggested by Referee #3. We believe the manuscript is much improved after addressing the comments of the three referees.

Our responses (black) follow each specific comment of the reviewers (blue) with any large changes to the text italicized and in quotations. Unless otherwise stated, the page and line numbers are in reference to the updated manuscript.

**Referee #1**

This study investigates the influence of springtime cyclones on surface ozone concentrations of two surface measurement sites in Ireland (Mace Head) and Portugal (Monte Velho). After a statistical evaluation of reanalysis data, the authors present four case studies – two for each measurement site – of cyclones leading to high ozone values at the surface. The results show that cyclones passing the measurement sites within a certain range can have a recognizable effect on the surface ozone concentration – either by transporting ozone-rich air directly from the stratosphere or by lower aged stratospheric air.
This manuscript is interesting and very carefully written and therefore in the scope of ACP.

**Major comments**

My only major comment deals with the cyclone intensity. Why did you choose the most intense cyclones?
**Authors' comments:** We choose the most intense cyclones for two reasons: a) so that the cyclones would be an exact match in both the ERA-Interim and the MACC data sets and moreover b) to ensure the airstreams controlling the ascent and descent were evident. Another benefit of selecting the most intense cyclones would be for storm compositing, as was done in Knowland et al. (2015). The most intense cyclones tend to follow similar storm track pathways within the main North Atlantic or North Pacific storm track region. Moderate to weak cyclones are not located solely in the major storm track pathways and tend to be shorter-lived (Knowland et al., 2015, ACP), resulting in a less coherent composite. The N1 cyclone was one of the cyclones in the NA composite in Knowland et al. (2015).

We have added the following text to Sect. 3.2.1 (Page 9, lines 19-21) to clarify this approach:

*"This ensures that the cyclones will be within the top 20 % of the cyclones in the NA and that the cyclone is an exact match identified in both the ERA-Interim and the MACC reanalysis data sets (not shown). Moreover, the airstreams within strong cyclones will be more evident than in the weaker cyclones (Catto et al., 2010)."*

Would it be better to investigate in addition also "normal" or "weaker" cyclones to get a broader picture of the influence of cyclones on surface ozone?  Do only strong cyclones have this impact on the ground stations?  Is there a clear relationship of high ozone and intense cyclones?  Can weaker cyclones have a stronger impact on the measured ozone by advecting aged low-level ozone-rich air to the measurement sites?

**Authors' comments:**  The statistics presented in Sect. 4.1 were performed on all tracks, not just the most intense cyclones.  For the reasons described above, we selected the most intense cyclones for the case studies.   As we learned from the case studies, the mean intensity of the cyclone is not the only factor determining the strength of the dry intrusion:  the S2 cyclone, which was weaker than the S1 and N2 cyclones based on the relative vorticity, was characterized by very strong descent.   We also note that the position of the cyclone and both upstream and downstream pressure systems as much as the intensity of the cyclone that determines the impact on ground stations.

We tested the suggestion on the case study criteria made by the referee and found that if instead we select the cyclones for case studies based on the highest O3 for tracks which had at least two time steps of >75th pc O3, then moderate to weak cyclones are selected.  Hence, weaker cyclones can also impact surface O3 at ground stations .

The following text has been added to the Discussion (Page 31, Lines 3-12):

*"The case study analysis focused on the strongest cyclones associated with high O3 at Mace Head and Monte Velho such that 1) the cyclone tracks would be exact matches in the MACC and ERA-Interim reanalyses (Knowland et al., 2015) and 2) airstreams would be most evident (Catto et al., 2010). An alternative method to select case study cyclones would be to start by selecting the tracks associated with the highest (or lowest) O3.  This leads to the selection of moderate to weak cyclones (which are generally not located solely in the major storm track pathways and which tend to be shorter-lived (Knowland et al., 2015)).  While the case study analysis presented here highlights the regional differences in the strongest springtime mid-latitude cyclones associated with high surface O3, the relationship between cyclones and surface O3 likely depends on a number of complex factors including cyclone location relative to the measurement station and upstream and downstream pressure systems, cyclone intensity, stratosphere-to-troposphere transport, and long-range transport of trace gases within the free troposphere and the boundary layer."*

**Technical corrections**

**Authors' comments:**  Changed (Page 5, Line 19).

Section 2.2.2 Please indicate already here the spatial resolution of the MACC data (it follows in Sec 3.1, but for completeness it would be nice to have it already in the data description part)

**Authors' comments:**  The details of the spatial resolution are now included in the sentence in Section 2.2.2 (Page 5, Lines 30-31):

*"The MACC reanalysis has the same spatial (TL255) and temporal (6-hourly) resolution as the ERA-Interim reanalysis."*

Page 9, line 14:  Are two time steps enough to justify the term "persistent"?
**Authors' comments:**  The US national air quality standard for O3 is based on the daily maximum 8-hour average O3 concentration.  Therefore, two or more 6-hourly time steps where high $O_3$ was present in the area could result in a daily maximum 8-hour average of $O_3$ exceeding an air quality standard.

Text was added to justify the criteria that the cyclones must have two or more 6-hourly time steps associated with high $O_3$ (Page 9, lines 23-25):

*"This selects the cyclones that are associated with elevated levels of $O_3$ at the monitoring station for a prolonged period of time which can be related to averaging periods typically used for policy metrics such as the daily maximum 8-hour average $O_3$."*

Figure 4:  something went completely wrong with the labelling (see screenshot) -> please check
Figure 11: a,b,c,d labels are missing.
**Authors' comments:**  Thank you for bringing this to our attention.  We will check the on-line proofs carefully.  Figures 4 and 11 have been corrected.

**Referee #2**

General Remarks

The topic of an elevated stratospheric ozone contribution at the surface has been in the focus of numerous recent studies driven by air-quality considerations.  In particular, it has been a mystery to me why Mace Head is reached by descending air masses, which has not been observed that frequently elsewhere.  The paper is well written and deserves publication, but I recommend considering a few amendments.

Details

1) Introduction: Consider mentioning the role of STT in some of the recent air-quality studies, such as:

   Lefohn et al., Atmos. Environ. 45 (2011), 4845-4857
   Langford et al., JGR 117, D00V06 (2012)
   M. Lin et al., JGR 117, D00V22 (2012)
   Lefohn et al., Atmos. Environ. 62 (2012), 646-656
   Ma et al., ACP 14 (2014), 5311-5325
   Dempsey, Atmos. Environ. 98 (2014), 111-122
   M. Lin et al., Nature Communications 2015
   Itoh and Narazaki, ACP 16 (2016), 6241-6261
   Langford et al., JGR 122 (2017), 1312-1337

   High-lying sites are ore affected by STT (western US, Alps (Zugspitze, Jungfraujoch), Tibet).
   **Authors' comments:**   We thank the referee for these additional references.  We have used these references in the following text and in the responses below.   The following text was added to the Introduction (Page 2, Lines 26-28):

   *"This can result in surface $O_3$ exceeding air quality standards, especially at high elevations such as in the western USA (Langford et al., 2009, 2015; Lefohn et al., 2009; Lin et al., 2012, 2014).* In particular, Lin et al. (2012) found that *in spring 2010* up to 60 % of total modeled surface O3 in the western USA during air quality exceedances could be attributed to stratospheric intrusions of O3…"

2) Introduction/Discussion:  The penetration of STT air into the PBL has been rarely observed (you cite Davis and Schuepbach (1994); some more words can be found in Eisele et al., J. Atmos. Sci. 56 (1999), 319-330).  The probability of entrainment into the PBL seems to be low, which suggests subsidence during night-time followed by day-time turbulent mixing (Eisele et al., 1999; Ott et al., JGR 121 (2016), 3687-3706; Langford et al., 2017) to be a more reasonable mechanism.  Is there anything different at the seaside (Cooper et al., JGR 100 (2005), D23310; Itoh and Narazaki (2016))?  You mention a fraction around 50 %.  Could this be related to the chance of night-time subsidence?

**Authors' comments:** This is a very good point. The following text regarding the ability for O3-rich air to be entrained into the PBL has been added (Page 2, Lines 33 – Page 3, Line 4):

*"Air from the free troposphere is largely limited to daytime entrainment into the lowest layer of the atmosphere as the planetary boundary layer (PBL) height increases (e.g., Itoh and Narazaki 2016; Ott et al., 2016); however, the ability for O3-rich air to reach the surface depends on a complex array of factors including the diurnal cycle (Itoh and Narazaki 2016; Langford et al., 2009, 2012; Ott et al., 2016) and the seasonal cycle of the PBL height (Langford et al., 2015, 2017), the presence of convective mixing (Eisele et al., 1999, Langford et al., 2017), and the elevation of the monitoring station, which if located within the free troposphere, especially with the night-time collapse of the PBL, can experience direct STT (Langford et al., 2015, 2017)."*

We have also added to the sentence describing why there is a springtime peak in the seasonal O3 cycle at marine, background stations (Page 3, Lines 21-28):

*"Also in springtime, there is a peak in the seasonal O3 cycle at marine, background stations including Mace Head (Simmonds et al., 1997, Wilson et al., 2012, Derwent et al., 2016); at such stations there is often a minimum in O3 in summer, as a result of enhanced O3 loss by increased water vapor within the stable marine boundary layer (MBL) (Ayer et al., 1992; Oltmans and Levy, 1994), in contrast to urban environments with the peak in O3 in summer associated with photochemical production. Parrish et al. (2016) and Derwent et al. (2016) demonstrate that the seasonal cycle of O3 at MBL stations, including Mace Head, can be reasonably represented by two harmonics of the seasonal cycle with the first explaining most of the seasonal variation, capturing the summertime O3 loss (and therefore a late winter/early spring maxima) and the spring maxima in O3 within the free troposphere which is entrained into the MBL."*

3) Also constituents from originally lofted Canadian fire plumes were observed at Mace Head (Forster et al., JGR 106 (2001), 22887-22906

**Authors' comments:** We thank the referee for this comment. According to Forster et al. (2001), carbon monoxide (CO) can be transported from the regions of biomass burning in Canada to Europe, which can lead to enhanced CO at Mace Head (58% above background for August 1998 fire season) as the plume passes over. The following text has been added to the Discussion section (Page 30, Lines 33—Page 31 Line 2):

*"Biomass burning plumes from as far away as Canada have reached Europe, leading to enhanced CO at Mace Head as the plume passes over (Forster et al., 2001). Within a fire plume, $O_3$ can be produced associated with the increased precursors and this has the potential to enhance European $O_3$, including at Mace Head."*

P. 4, line 28: The spatial resolution seems to be sufficient to resolve STT layers. This is an important issue (e.g., Roelofs et al., J. Geophys. Res. **108** (2003), 8529; Eastham and Jacob, ACP 17 (2017), 2543-2553). However, there is, still, the issue of mixing even in high-resolution

models that can lead to a loss of information (two rather bad comparisons between measurements and a high-resolution ECMWF model results: Trickl et al., ACP 16 (2016), 8791-8815). This topic should be addressed somewhere in the paper.

**Authors' comments:** This is a very important distinction to make. We have added the following text to the end of Sect. 2.2 (Page 5, Lines 4-8):

*"While models and reanalyses with coarse horizontal resolution (> 100km) are able to identify stratospheric intrusions (Roelofs et al., 2003; Lin et al., 2015; Ott et al., 2016), the fine-scale nature of the $O_3$ filaments is best identified at higher horizontal resolutions (Buker et al., 2005; Lin et al., 2012; Ott et al., 2016), such as the MACC and MERRA-2 reanalyses and by the stratospheric tracer. However, even high-resolution reanalyses may struggle to represent the complex structure of stratospheric intrusions (e.g., Trickl et al., 2016)."*

**Referee #3**

**General Remarks**

This manuscript investigates how passing mid-latitude cyclones influence the surface ozone levels at two stations in Western Europe (Mace Head and Monte Velho) by comparing ozone observations and reanalysis data from ERA-Interim, MACC and MERRA-2. Four case studies were presented to support the main results. The strong impact of the downward transport of O3-rich air masses from the stratosphere on the surface ozone levels was pointed out.

The topic and the general length of the paper are relevant for publication in ACP. Generally the paper is compact and well-written, however Sec. 4.2.1-4.2.3 should be improved as suggested below. The structure of the paper is logical. The introductory discussion is adequate and the referencing is sufficient. In a few cases more details on the methods are needed. Some of the figures need improvement.

For the reasons mentioned above and below the paper is appropriate for publication in ACP after a major revision.

**Specific comments:**
**Authors' comments:** We want to thank the referee for the many constructive comments regarding the results section of this manuscript. We have made modifications to the text and figures in Sect. 4.2.1-4.2.4 to improve the readability of the figures and the discussion.

-Shorten Sect. 4.2.1 and improve the text of this section (write more compact and structured). Perhaps add a table to shorten the text.
**Authors' comments:** Section 4.2.1 describes the first case study, the N1 cyclone. We have reduced the text in this section by modifying the text describing Fig. 6 to focus on the airstreams; instead of adding a table to shorten the text, we have removed details already discussed in relation to Figs. 3 and 4 in the geographical coordinate system  Now the paragraphs describing Fig. 6 have been reduced to one paragraph instead of three (Page 17 Line 26—Page 18, Line 5).

This section does include necessary text as to how to read the cyclone-centered figures, especially the TFP figures. We believe it would make it more difficult to read these figures without this text unless the reader was familiar with them from previous literature.

-Section 4.2.2 should be better weighted (better incorporate the amount of text compared to the number of figures).
**Authors' comments:** To better weight text and figures in Sect. 4.2.2, Fig. 8 has been reduced from four panels to two panels for the initial description of the S1 cyclone using the geographical coordinates (Pages 21-22).

Also, the text in Sect. 4.2.2 has been modified to focus on the residual O3 that becomes entrained into the cyclone, which was not seen in the N1 cyclone in the same way, and the differences in the S1 cyclone using the two reanalyses.

-In comparison Section 4.2.3 is extremely brief and all information is shifted to the Supplement. -> I get the general impression that Sect. 4.2.1-4.2.3 must be better weighted and the text must be improved and more compact written.  Sometimes hard to follow for the reader.

**Authors' comments:**
We appreciate this feedback and in the Sect. 4.2.2-4.2.4 the emphasis in the text is now on how the remaining case studies are different to the N1 cyclone and therefore provide insight on how strong cyclones can bring high levels of O3 to Mace Head and Monte Velho.

Specifically for Sect. 4.2.3, we have added a new figure and text (inserted below; Pages 26 Line 13 – Page 27 Line 3).  The new Fig. 12 highlights the synoptic conditions and O3 distribution at the surface over the North Atlantic sector during the life time of the N2 cyclone (based on the bottom panel of the N2 figures in the supplemental).

[Figure]

**New Figure 12.  Synoptic conditions and O3 distribution for the N2 cyclone at 1000 hPa:  MACC O3 (color) and horizontal wind vectors (10 m s$^{-1}$ reference arrow) and MSLP (solid contours, 5 hPa intervals) on a) 18 May 2006 18 UTC, b) 21 May 2006 06 UTC and c) 22 May 2006 06 UTC.  The approximate cold front (b,c; blue line) and Monte Velho (pink open circle) are included.**

The text in Section 4.2.3 provides an overview of the N2 cyclone, however the main discussion for the N2 cyclone is still in the supplemental material since the N2 cyclone is similar to the N1 and S1 cyclones discussed in Sects. 4.2.2 and 4.2.3, respectively.  The following text is Section 4.2.3 on Page 26 (new text in italics):

*"The N2 cyclone is a strong cyclone which passed north of Monte Velho and advected high O3 to the observation site. Cyclogenesis of the N2 cyclone occurs off the east coast of Florida on 17 May 2006 at 12 UTC (not shown) and then tracks northeastward (Fig. 2b).  The synoptic conditions at the time of the N2 cyclone are similar to the N1 cyclone:  About a day after cyclogenesis, the N2 cyclone is located just south of Newfoundland and downwind of the* N2 cyclone are a series of low pressure systems across the NA region as well as a strong Azores High stretching over most of the NA *(Figs. 12a and S7c).  The large pressure gradient results in strong westerly winds at near 45° N (Fig. 12a) throughout the troposphere (Fig. S7). Similar to the N1 cyclone, stratospheric O3 descends into the troposphere within the low pressure systems in the NA ahead of the N2 cyclone (Figs. 12a and S7).  Similar to the S1 cyclone, a high pressure system develops to the west of the N2 cyclone (Fig. 12b-c), supporting upper-level descent*

*behind the N2 cyclone's cold front and negating transport across the NA from North America (Figs. 12b-c, S8 and S9).*

*As was seen for the N1 cyclone, the N2 cyclone is also associated with low O3 before high O3 is observed at the monitoring station due to the passage of a cold front (Fig. 12b-c).* Since the N2 cyclone is similar to the N1 *and S1 cyclones discussed in Sects 4.2.2 and 4.2.3, respectively, full details* for this case can be found in the Supplemental material (Sect. S2.3).*"*

-Sometimes the numbers for statistics are rather low (number of strong cyclones, see minor comments).  Comment on this!
**Authors' comments:**  All cyclones, not only the strong cyclones, were used in the statistics presented in Sects. 4.1. To clarify, we have added the following (italics, Page 11, Lines 7-8):

"*Here, tracks were not filtered for intensity.  For springtime during the period of observations from 1988—2010 at Mace Head, the average number of the total cyclone tracks associated with O3 > 75th pc ….as a percentage of the total tracks in each region.*"

The intensity of the cyclone was only a factor in selecting the case study cyclones to ensure that 1) the cyclones are an exact match in ERA-Interim and the MACC reanalysis and 2) the airstreams within strong cyclones will be more evident than in weaker cyclones.  This additional information regarding the case study cyclone identification methodology has been added to the manuscript (Page 9, Lines 18-20) following Referee #1's comment.

-Perhaps it would also be useful to show observed ozone time series for the whole months of your case studies?
**Authors' comments:**  We do not believe a time series of the observed O3 alone would add further value to the manuscript.  For example, the Figure 1 below is the Mace Head hourly O3 observed (dashed blue contour) and the 6-hourly averaged O3 (thick black contour) for March 2007 when the N1 cyclone occurred.  While the observed change from low to high O3 can be seen in the time series at the time of the N1 cyclone, it is hard to pick out the impact of the cyclone on the O3 variability in the time series without prior knowledge of the cyclone.   The ozone time series shows strong variability especially from the diurnal and seasonal cycles.

[Figure]

**Figure 1 - Time series of 6-hourly averaged O3 (thick black contour) calculated from the 1-hour observed O3 (blue dashed contour) for Mace Head for March 2007. The N1 cyclone reached maximum vorticity on 4 March 2007 12 UTC (black diamond).**

However, figures showing the O3 time series with additional information such as departure from the seasonal mean, variability explained by the diurnal cycle, possible outside influences on surface O3 such as long-range transport of emissions precursors (i.e., biomass burning events as suggested by Referee #2), long-range transport of O3, stratospheric O3, and frequency of storms in the area could be enlightening, although difficult to do without extensive further analysis and model simulations.

Your results must be very sensitive to the vertical distribution of ozone in the tropopause layer (reanalysis data). Have you investigated this in detail? How accurate are the reanalysis data at these altitudes?

**Authors' comments:** The distribution of O3 in the upper troposphere and lower stratosphere is sensitive to the representation in the reanalyses. According to Innes et al. (2008), the MACC reanalysis biases compared to ozonesondes are within ± 5 to 10% in the Northern Hemisphere between 200 and 1000 hPa, and less than 10% over Europe and North America when compared to MOZAIC flight data. Wargan et al. (2017) recommends that studies of mid-latitude stratosphere-troposphere exchange using the MERRA-2 reanalysis should limit the time period to when the EOS Aura data was assimilated (October 2004 until present). Specifically, over Europe (45°-60°N, 0°-60°E), the springtime MERRA-2 ozone profiles in 2005 resemble the vertical structure of the ozonesondes, although biased low, with $r^2 > 0.6$ down to 5km below the tropopause (Wargan et al., 2017). In MERRA-2, a low bias in the UT is expected as the production of O3 from lightning-$NO_x$ and pollution (including surface emission sources of O3 precursors) is not included (Wargan et al., 2015). The GEOS-5 model has a higher vertical grid spacing of about 1 km in the UTLS compared to the MLS data (2.5 – 3 km) (Wargan et al., 2015).

We have added discussion of the representation of UTLS O3 in reanalyses to Sect. 4.2.2 (Page 22, Line 13 – Page 23 Line 6):

*"The vertical transport of O3 will be sensitive to the strong gradients of O3 near the tropopause...Both the MACC and MERRA-2 O3 have good agreement to independent*

*observations in the upper troposphere/lower stratosphere (UTLS) region (Inness et al., 2013; Wargan et al., 2015, 2017). In the Northern Hemisphere (NH) extra-tropics, MACC O3 between 500 and 100 hPa has a bias of -10--0\% when compared to ozonesondes profiles (Inness et al., 2013) and Wargan et al. (2017) found that the average MERRA-2 O3 profile over Europe (45°-60°N, 0°-60°E) in the spring of 2005 (after the assimilation of MLS and OMI satellite retrievals) captures the vertical structure of the ozonesondes in that region ($r^2$ > 0.6), although biased low (up to -20 %) in the UTLS."*

**Minor comments and technical corrections:**

Page 2, line 7: Add also the mean latitude for the Icelandic low, since you also give this for the Azores High (line 9).
**Authors comment:** The mean latitude added (Page 2, Line 7)

Page 2: Add this reference to your discussion: Trickl, T., O.R. Cooper, H. Eisele, P. James, R. Mucke, and A. Stohl (2003), Intercontinental transport and its influence on the ozone concentrations over central Europe: Three case studies, J. Geophys. Res., 108, 8530, doi: 10:1029/2002JD002735, D12.
**Authors' comments:** Reference added to Introduction (Page 2, Lines 17-21)

Page 3, line 25-30: What kind of instrument was used for the ozone measurements?
**Authors' comments:** Details on the instruments has been added to Sect. 2.1 (Page 4, Line 7 and 12-14)

Page 3, line 29: Replace "pink" by "orange"
**Authors' comment:** Text modified as suggested.

Page 4, line 5: More clearly point out the difference between the present paper and Knowland et al., 2015.
**Authors' comment:** Knowland et al., 2015 identified and composited together the most intense cyclones over the North Pacific and North Atlantic sectors. In this paper, we are looking at a few individual cyclone cases over the North Atlantic/European sector.

We have added to the text (italics; Page 4, Lines 21-22):

*"Through a composite cyclone analysis,* it was clearly shown *that intense cyclones over the North Pacific (NP) and NA regions* redistribute O3 and CO horizontally and vertically within the WCB and DI airstreams (Knowland et al., 2015)."

Page 4, line 12-13: Add webpages for ERA-Interim, MACC, and MERRA-2 reanalysis.
**Authors' comment:** Webpages added to the "Data availability" at end of paper (Page 33).

Page 4, line 25: What does "cycle 31r2" mean?

**Authors' comment:** Added following text to clarifying that "cycle" refers to the IFS model version (Page 5, Lines 11-13):

*"….with the ECMWF's Integrated Forecast System (IFS) cycle 31r2 (a full history of changes made to the ECMWF IFS, as indicated by the cycle number, is available at https://www.ecmwf.int/en/forecasts/documentation-and-support/changes-ecmwf-model), with a sequential…."*

Page 5, line 2: What does "cycle 36r1" mean?
**Authors' comments:** Added text to clarify the IFS model version used in MACC is more recent version compared to ERA-Interim (Page 5, Lines 23-24).

Page 5, line 13: What is the explanation for the underestimate?
**Authors' comments:** It is not known why there is an underestimation in the MACC reanalysis O3 during spring over central and northern Europe (Innes et al., 2013). We have added the following text to provide possible explanations based on Innes et al. (2013) (Page 6, Line 3-8):

*"The reason(s) for the biases in the seasonal O3 is not known although Innes et al. (2013) hypothesize that the biases in the MACC surface $O_3$ are related to the diurnal cycle possibly due to 1) there is no diurnal cycle in the $NO_x$ emissions used in the CTM resulting in negative O3 biases during the day-time and positive O3 biases during the night-time, 2) misrepresentation of vertical mixing between the boundary layer and free troposphere, and/or 3) less observations are available (and therefore not assimilated) at night."*

Page 6, line 2-5: Reorder the references in chronological order (and for same years in alphabetical order).
**Authors' comments:** Thank you for bringing this to our attention. Reference order changed.

Page 6, line 11: Write out STFR.
**Authors' comments:** Acronym now defined.

Page 6, line 19: Reorder the references in chronological order.
**Authors' comments:** Reference order changed.

Page 6, line 27: "< 1000 km" in what time?
**Authors' comments:** This distance would be for the life cycle of the cyclone. Sentence now reads (Page 7, Lines 20-21):

*"On completion of the tracking process, the stationary cyclones (travel < 1000 km during the cyclone's life cycle) and short-lived cyclones (cyclone's life cycle lasts < two days)…."*

Page 8, Figure 1: The letter size of lat and lon numbers is too large compared to the track density numbers.
**Authors' comments:** The font size in Fig. 1 has been improved as per suggestion (Page 8).

Page 9, Table 1: The number of strong cyclones used in the study is very low. Is this number high enough for your statistics?

**Authors' comments:** All cyclones, not only the strong cyclones, were used in the statistics presented in Sects. 4.1. The number of strong cyclones presented in Table 1 was used in the selection of the case study cyclones to ensure the airstreams within the case study cyclones were identifiable.

Page 10, line 23: What does "SGF" mean?

**Authors' comments:** Abbreviation now defined in main text (Page 11, Line 12).

Page 11, Table 2: Last column (right): "percent" -> "Percent"

**Authors' comments:** Text has been changed.

Page 11, Table 2: The number of years given in the brackets (SGF) is very low. Can you comment on this low number for your statistics?

**Authors' comments:** Nearly all the years with more tracks associated with high O3 for the North region associated with Mace Head are significant (15 of the 18 years) and half of the years with more tracks associated with low O3 for the South region associated with Mace Head are significant (8 of the 16). It is true there are a small number of years which had significant differences for tracks passing in the other regions near Mace Head and in the regions near Monte Velho. The low number of significantly different years is likely because as we move away from the NA storm track region more tracks are associated with both high and low O3 so it is expected the number of years with more tracks associated with high O3 or with low O3 would be less and therefore have less significant differences. This provided the motivation to look at the case study cyclones.

We have modified the text to read (Page 13, line 7-10):

"The number of these years having more tracks in the North, Center, and South regions associated with high O3 is almost equal to the number of years which had more tracks associated with low O3. Very few of the years have a significant $\chi^2$ difference, possibly due to the percent of tracks associated with both high and low O3 increases (up to 31 % of tracks) the further away from the main NA storm track (Table 2)."

Page 14, Figure 3 legend: Always mention latitude first and longitude. Check this through the paper.

**Authors' comments:** The order of the coordinates has been corrected in the figure captions for Figs. 3, 4, 8, 9, 12, and 13 as well as in the text in Sect. 4.2.2.

Page 15, Figure 4: It is very hard to recognize Ireland in the maps (especially in b and c).

**Authors' comments:** In order to improve the readability of Fig. 4, the spacing between wind bars has been increased, the reference arrow in Fig. 4d has been increased from 10 to 20 m s$^{-1}$,

and the ascent contour interval has increased from 10 to 15 hPa h$^{-1}$. By reducing the amount of information, the country borders are now more visible.

**Authors' comments:** To be consistent with the other figures, the 500 hPa (Fig. 6a) is above the 1000 hPa (Fig. 6c), however in the manuscript it is more logical to describe the surface front features prior to the upper-level features in Fig. 6. We have added the following sentence to the start of the paragraph, prior to discussing Fig. 6c as the cold frontal boundary is a feature throughout all four panels in Fig. 6 (Page 17, Lines 26-27):

*"The main feature at 4 March 2007 00UTC in both the meteorological fields and O3 distribution is the cold frontal boundary which separates the WCB and DI airstreams (Fig. 6). The strong cold frontal boundary features as the sharp gradient in the 1000 hPa O3 distribution and the curve in the isobars in the southwest quadrant of Fig. 6c (similar to Fig. 3). The front is identified …..Fig. 6b). "*

**Authors' comments:** The font size of the panel lettering has been reduced in all figures (including the figures in the supplemental material).

**Authors' comments:** The thickness of the white lines in Fig. 6a has been increased (Page 19). We have increased the thickness of the white lines in Figs. 7a, 10a, and 11a.

**Authors' comments:** This sentence has been shortened to focus on the main point of the sentence, that the MERRA-2 O3 shows elevated O3 within the dry airstream in same location as seen in the MACC reanalysis O3 (Page 21, Lines 7-9) and now reads:

*"In the troposphere, MERRA-2 O3 is expected to be biased low (Sect. 2.2.3), and while the MERRA-2 troposphere O3 mixing ratios are lower in the N1 cyclone compared to the MACC reanalysis, there is still relatively higher O3 within the DI airstream reaching the surface (Figs. S1b,d and S3b-d) as seen in Figs. 6 and 7."*

**Authors' comments:** Changing the thickness of the contours or changing the style of the contours did not improve the readability of Fig. 9. Instead, we decided to add an inset of the 15° by 15° region over Ireland and the UK, without the wind arrows, in order to improve the readability of the white contours over Mace Head (Page 23).

Page 21-26: In this section you present many figures, however very little text to the figures. Weight figures and text better.

**Authors' comments:**  Changes made to Section 4.2.2 have been addressed in the Specific Comments above.  Specifically, the number of panels in Fig. 8 has been reduced to two, having removed the panels that are not necessary for the text.  The text has also been modified to focus on the differences in the S1 cyclone compared to the N1 cyclone.

Page 31, line 15-16:  This sentence is somehow stand-alone.  What is the message?
**Authors' comments:**  The message is that the presence of high $O_3$ in the mid-troposphere in the composite cyclone of Knowland et al. (2015) is related to a parent low or downstream cyclone.  We have modified the text to make a clearer connection between the composite analysis of Knowland et al. (2015) and the case study cyclones.   The final sentence has been modified to read (Page 31, Line 35 – Page 32, Line 5):

*"In addition, the case studies depict elevated $O_3$ from either a parent low pressure system (e.g., the N1 cyclone case study) or an upstream decaying cyclone (e.g., the S1 and N2 cyclones) can become entrained within the cyclone; this is a strong feature in mid-latitude cyclones as there is elevated $O_3$ (up to 5 ppbv greater than the background composite) similarly to the north of the NA composite cyclone center, not within the main DI airstream and separated from the $O_3$-poor WCB, that wraps cyclonically around the composite cyclone (Knowland et al., 2015)."*

Page 31, line 18:  "We have shown passing cyclones have" -> "We have shown that passing cyclones have"
**Authors' comments:**  Text changed as per suggestion.

Page 31, line 26:  For a better structure after "Specifically:"  Add a "-" to all small paragraphs that follow and use an insertion slide-in.
**Authors' comments:**  Text changed as per suggestion.

Supplement:  Figs. S2c, S4c, and S6c are not very useful, since they are too dark.
**Authors' comments:**  The STFR color bar has been changed to use blue color scale, with low STFR as light blue and increasing to a dark blue, but never as dark as the original blues in the Figs. S2, S4, and S6.